# Intrinsically disordered protein PID-2 modulates Z granules and is required for heritable piRNA-induced silencing in the *Caenorhabditis elegans* embryo

Maria Placentino[1,2], António Miguel de Jesus Domingues[1] (iD), Jan Schreier[1,2] (iD), Sabrina Dietz[2,3],
Svenja Hellmann[1], Bruno FM de Albuquerque[1,4] (iD), Falk Butter[3] (iD) & René F Ketting[1,5,*] (iD)

## Abstract

In *Caenorhabditis elegans*, the piRNA (21U RNA) pathway is required to establish proper gene regulation and an immortal germline. To achieve this, PRG-1-bound 21U RNAs trigger silencing mechanisms mediated by RNA-dependent RNA polymerase (RdRP)-synthetized 22G RNAs. This silencing can become PRG-1-independent and heritable over many generations, a state termed RNA-induced epigenetic gene silencing (RNAe). How and when RNAe is established, and how it is maintained, is not known. We show that maternally provided 21U RNAs can be sufficient for triggering RNAe in embryos. Additionally, we identify PID-2, a protein containing intrinsically disordered regions (IDRs), as a factor required for establishing and maintaining RNAe. PID-2 interacts with two newly identified and partially redundant eTudor domain-containing proteins, PID-4 and PID-5. PID-5 has an additional domain related to the X-prolyl aminopeptidase APP-1, and binds APP-1, implicating potential N-terminal proteolysis in RNAe. All three proteins are required for germline immortality, localize to perinuclear foci, affect size and appearance of RNA inheritance-linked Z granules, and are required for balancing of 22G RNA populations. Overall, our study identifies three new proteins with crucial functions in *C. elegans* small RNA silencing.

**Keywords** PID-2; PID-4; PID-5; piRNA; Z granule
**Subject Categories** Development; RNA Biology
**The EMBO Journal (2021) 40: e105280**

## Introduction

Germ cells are responsible for transmitting genetic information to the next generation. Therefore, genome stability should be tightly controlled in these cells. The integrity of the genome is constantly threatened not only by external factors, such as irradiation and mutagenic agents, but also by intrinsic factors resident in the genome, such as transposable elements (TEs). Consequently, organisms have evolved a variety of mechanisms to counteract these threats. Among these, small RNA pathways often play important roles in controlling TE activity. In many animals, TEs are recognized and silenced in the germline by a specific small RNA pathway: the Piwi pathway. Piwi proteins represent a specific subclade of Argonaute proteins that exert their silencing function upon loading with their cognate small RNA, named piRNA (Piwi-interacting RNA), that specifies the target transcript. The Piwi/piRNA pathway is essential in most organisms for TE silencing, but also TE-unrelated effects have been well-described (Ghildiyal & Zamore, 2009; Malone & Hannon, 2009; Ketting, 2011; Siomi *et al*, 2011; Ozata *et al*, 2019).

The main and likely only active Piwi protein of *Caenorhabditis elegans* is PRG-1; it binds to piRNAs, which in *C. elegans* are named 21U RNAs, to form a silencing complex. In contrast to other organisms, loss of the PRG-1/21U RNA pathway in *C. elegans* causes the reactivation of only a limited set of transposable elements, for instance Tc3 (Das *et al*, 2008), and does not cause immediate sterility (Cox *et al*, 1998; Batista *et al*, 2008; Das *et al*, 2008; Wang & Reinke, 2008), even though germ cells are progressively lost over generations (mortal germline phenotype, Mrt) (Simon *et al*, 2014). The discrepancy between the Piwi-mutant phenotypes observed in *C. elegans* and other animals can be explained by the fact that PRG-1 initiates a silencing response that is executed by a different set of Argonaute proteins—the worm-specific Argonaute proteins

1   Biology of Non-coding RNA Group, Institute of Molecular Biology (IMB), Mainz, Germany
2   International PhD Programme on Gene Regulation, Epigenetics & Genome Stability, Mainz, Germany
3   Quantitative Proteomics Group, Institute of Molecular Biology (IMB), Mainz, Germany
4   Graduate Program in Areas of Basic and Applied Biology, University of Porto, Porto, Portugal
5   Institute of Developmental Biology and Neurobiology, Johannses Gutenberg University, Mainz, Germany
    *Corresponding author. Tel: +49 6131 3921470; E-mail: r.ketting@imb.de

(WAGOs)—while in other studied model systems this does not happen. Upon target recognition by PRG-1, an RNA-dependent RNA polymerase (RdRP) is recruited to the targeted transcript, which is used as a template for the synthesis of complementary small RNAs, named 22G RNAs. For this step, the RdRP RRF-1 is required, as well as so-called Mutator proteins (Zhang *et al*, 2011; Phillips *et al*, 2012; Phillips *et al*, 2014). The 22G RNAs, characterized by the 5′ triphosphate group resulting from the RdRP-driven synthesis, are loaded onto WAGO proteins, such as HRDE-1 and WAGO-1 (Gu *et al*, 2009; Ashe *et al*, 2012; Buckley *et al*, 2012; Shirayama *et al*, 2012), that reinforce the silencing started by PRG-1. Occasionally, in a seemingly stochastic and poorly understood manner, this silencing can become independent of PRG-1 itself and self-sustainable. This form of silencing is extremely stable and can be maintained across many generations in the absence of PRG-1. It is characterized by the deposition of heterochromatic marks at the targeted locus, depends on HRDE-1 and Mutator activity, and it is known as RNAe (RNA-induced epigenetic gene silencing) (Ashe *et al*, 2012; Luteijn *et al*, 2012; Shirayama *et al*, 2012). RNAe can thus explain why transposons remain silenced in the absence of PRG-1. Indeed, in *prg-1; hrde-1* double mutants, lacking both 21U RNAs and RNAe, the activity of Tc1 transposons increases to levels comparable to Mutator mutants, indicating that HRDE-1 activity is sufficient to maintain Tc1 silencing in *prg-1* mutants (de Albuquerque *et al*, 2015).

PRG-1/21U RNA complexes can recognize a target transcript via imperfect base-pair complementarity, allowing up to four mismatches (Bagijn *et al*, 2012; Lee *et al*, 2012). As a consequence of this mismatch tolerance, PRG-1 is potentially able to recognize and silence many different sequences, including endogenous genes (Bagijn *et al*, 2012; Gu *et al*, 2012). Another small RNA pathway, guided by 22G RNAs bound to the Argonaute protein CSR-1, has been implicated in counteracting such PRG-1-mediated silencing of genes that should be expressed (Claycomb *et al*, 2009; Gu *et al*, 2009; Lee *et al*, 2012; Shirayama *et al*, 2012; Conine *et al*, 2013; Seth *et al*, 2013; Wedeles *et al*, 2013; Shen *et al*, 2018). CSR-1-bound 22G RNAs are made by the RdRP EGO-1 in a mostly Mutator-independent manner (Claycomb *et al*, 2009; Gu *et al*, 2009). Interestingly, an opposite scenario has also been described: PRG-1 has been shown to direct Mutator activity to non-CSR-1 targets in embryos that set up a 22G RNA silencing response *de novo* (de Albuquerque *et al*, 2015; Phillips *et al*, 2015). These seemingly contradictory findings—CSR-1 counteracting inappropriate PRG-1 targeting versus PRG-1 directing Mutator activity away from CSR-1 targets—may be explained by considering that two different developmental stages have been analysed to arrive at the proposed models. The protective role of CSR-1 has been seen in the adult germline (Claycomb *et al*, 2009; Gu *et al*, 2009; Lee *et al*, 2012; Shirayama *et al*, 2012; Conine *et al*, 2013; Seth *et al*, 2013; Wedeles *et al*, 2013; Shen *et al*, 2018), whereas the protective role of PRG-1 likely operates in embryos (de Albuquerque *et al*, 2015; Phillips *et al*, 2015). Possibly, PRG-1 has different modes of actions at these two developmental stages. Another result that indicates differential PRG-1 activities in adults versus embryos comes from studies on HENN-1, the enzyme that 2′-O-methylates 21U RNAs. In adults, 21U RNA levels are not affected by loss of HENN-1 (Kamminga *et al*, 2012), while in embryos 21U RNAs are reduced in *henn-1* mutants (Billi *et al*, 2012; Montgomery *et al*, 2012). Given that 2′-O-methylation has been shown to stabilize small RNAs, in particular when they base pair extensively to their

targets (Ameres *et al*, 2010), it is feasible that PRG-1 recognizes targets with near-perfect complementarity to its cognate 21U RNA only in the embryo and employs more relaxed 21U RNA targeting in the adult germline. Indeed, maternally provided PRG-1 protein is required to establish PRG-1-driven silencing of a 21U RNA sensor transgene that has perfect 21U RNA homology, suggesting that this silencing is set up during early development, and not in the adult germline (de Albuquerque *et al*, 2014). Whether the maternal contribution of PRG-1 is sufficient to induce silencing has not been tested thus far.

A third small RNA pathway is driven by so-called 26G RNAs (Yigit *et al*, 2006; Han *et al*, 2009; Conine *et al*, 2010; Billi *et al*, 2014). These are made by the RdRP enzyme RRF-3, which acts in a large protein complex containing well-conserved proteins such as Dicer, GTSF-1 and ERI-1 (Kennedy *et al*, 2004; Duchaine *et al*, 2006; Thivierge *et al*, 2012; Billi *et al*, 2014; Almeida *et al*, 2018). These 26G RNAs can be bound by the Argonaute protein ERGO-1, or by two closely related paralogs, the Argonaute proteins ALG-3 and ALG-4 (ALG-3/-4). ERGO-1 mostly targets transcripts in the female germline and the early embryo, and is required to load the somatic, nuclear Argonaute protein NRDE-3 with 22G RNAs (Han *et al*, 2009; Gent *et al*, 2010; Vasale *et al*, 2010; Billi *et al*, 2014; Almeida *et al*, 2019a). The 26G RNAs bound by ERGO-1 require HENN-1-mediated 2′-O-methylation in both the adult germline and the embryo (Billi *et al*, 2012; Montgomery *et al*, 2012; Kamminga *et al*, 2012). ALG-3/-4-bound 26G RNAs are not modified by HENN-1 (Billi *et al*, 2012; Montgomery *et al*, 2012; Kamminga *et al*, 2012) and are specifically expressed in the male gonad (Han *et al*, 2009; Conine *et al*, 2010; Conine *et al*, 2013).

Many of the above-mentioned proteins are found in phase-separated structures, often referred to as granules or foci. Mutator proteins that make 22G RNAs are found in so-called Mutator foci, whose formations is driven by MUT-16, a protein with many intrinsically disordered regions (IDRs) (Phillips *et al*, 2012; Uebel *et al*, 2018). The RdRP EGO-1, as well as the Argonaute proteins CSR-1, PRG-1 and a number of others, are found in P granules (Batista *et al*, 2008; Wang & Reinke, 2008; Claycomb *et al*, 2009; Updike & Strome, 2010), characterized by IDR proteins such as PGL-1 (Kawasaki *et al*, 1998) and DEPS-1 (Spike *et al*, 2008), which are also required for P granule formation. Finally, Z granules are marked by the conserved helicase ZNFX-1 and the Argonaute protein WAGO-4 (Ishidate *et al*, 2018; Wan *et al*, 2018). Z granules are related to the inheritance of small RNA-driven responses via the oocyte (Ishidate *et al*, 2018; Wan *et al*, 2018) and are typically found adjacent to P granules. However, in primordial blastomeres, Z and P granules appear to be merged (Wan *et al*, 2018). For Z granules, no IDR protein that may drive their formation has been identified yet. The function of ZNFX-1 is also not clear, but it has been demonstrated that it interacts with the RdRP EGO-1 and that it is required to maintain the production of 22G RNAs from the complete length of the targeted transcript (Ishidate *et al*, 2018). In the absence of ZNFX-1, relatively more 22G RNAs are found to originate from the 5′ part of the RdRP substrate, suggesting that ZNFX-1 may have a role in maintaining or relocating the RdRP activity to the 3′ end of the substrate. Despite the fact that material exchange between these three types of structures (P, Z granules and Mutator foci) seems obvious, how this may happen is currently unknown.

Here, we describe the characterization of a novel gene, *pid-2*, which we identified from our published "piRNA-induced silencing

   

defective" (Pid) screen (de Albuquerque *et al*, 2014). Our analyses show that the IDR protein PID-2 is essential for initiation of silencing by maternally provided PRG-1 activity. However, PID-2 is also required for efficient maintenance of RNAe and shows defects in many different small RNA populations indicating that PID-2 does not only act together with PRG-1. Interestingly, we noticed a drop of 22G RNA coverage specifically at the 5′ end of RRF-1 substrates, suggesting that PID-2 may be involved in stimulating RdRP activity or processivity. At the subcellular level, PID-2 is found in granules right next to P granules, and the absence of PID-2 affects size and number of Z granules, suggesting that PID-2 itself may also be in Z granules. We also identify two PID-2-interacting proteins, PID-4 and PID-5, with an extended Tudor (eTudor) domain. In addition, PID-5 has a domain that closely resembles the catalytic domain of the X-prolyl aminopeptidase protein APP-1. Loss of both PID-4 and PID-5 phenocopies *pid-2* mutants in many aspects, including the effects on small RNA populations and on Z granules. At steady state, both PID-4 and PID-5 are themselves mostly detected close to or within P granules. We hypothesize that the here identified PID-2/-4/-5 proteins have a role in controlling RdRP activity, and do so by affecting protein and/or RNA exchange between different germ granules.

# Results

## PID-2 is an IDR-containing protein required for 21U RNA-driven silencing

We have previously performed and published a forward mutagenesis screen in which we identified several mutants that are defective for 21U RNA-driven silencing (piRNA-induced silencing defective: Pid) (de Albuquerque *et al*, 2014). In this screen, the de-silencing of a fluorescent 21U RNA target was used as read-out. The silencing of the transgene depended on both PRG-1 and 22G RNAs (Bagijn *et al*, 2012), allowing for the isolation of mutants that affect 21U or 22G RNAs; we will refer to this PRG-1-dependent state as 21U sensor(+). Here, we focused our attention on a mutant, defined by the allele *xf23*, resulting in a point mutation (tgg → tga) that causes a premature stop codon (W122X) in the gene Y48G1C.1. This gene encodes a protein with disordered N- and C-terminal regions (Fig 1A). The rest of the encoded protein is more structured (Fig 1A), even though no predicted domains were detected. We also obtained a publicly available deletion allele of Y48G1C.1, *tm1614* (Fig 1A; Barstead *et al*, 2012). Imaging revealed that animals homozygous for *xf23* or *tm1614* showed a strongly penetrant silencing defect of the 21U sensor(+), even though the defect is less severe compared with Mutator mutants (Fig 1B). Quantification of the de-silencing induced by both alleles using RT–qPCR revealed 10–20% activation of the 21U sensor(+) compared with Mutator mutants (Fig EV1A). A single-copy transgene expressing 3xFLAG-tagged Y48G1C.1, and to a lesser extent GFP-tagged Y48G1C.1, driven by its endogenous promoter and 3′ UTR could rescue the 21U sensor(+) phenotype (Fig EV1B–E). We conclude that the mutation in Y48G1C.1 plays a role in 21U sensor(+) silencing, and named the gene *pid-2*.

The 21U sensor can also be in a state of RNAe: 21U sensor (RNAe). In this state, its silencing no longer depends on PRG-1, but does still rely on 22G RNAs (Ashe *et al*, 2012; Luteijn *et al*, 2012;

Shirayama *et al*, 2012). In contrast to the sensor(+) reactivation experiment, most *pid-2* mutant animals did not reactivate the 21U sensor (RNAe) (Fig 1B). Nonetheless, we did detect reactivation of the 21U sensor (RNAe) in some animals, most notably in *pid-2* (*xf23*) mutants (Fig 1B). Continuous culturing of independent cultures confirmed recurrent loss of RNAe status in *pid-2*(*xf23*) mutants, particularly at elevated temperature (Fig EV1F). Such loss of RNAe was much less frequent in *pid-2*(*tm1614*) animals (Fig EV1F). Given that the reactivation of the sensor(+) was also less effective in *pid-2*(*tm1614*) mutants (Fig 1B), we assume that *pid-2*(*tm1614*) is a weaker allele than *pid-2*(*xf23*), and as such only has a very weak phenotype in the more stringent sensor (RNAe) assay, while it has an easily scored phenotype in the sensor(+) assay. RT–qPCR showed that *pid-2*-mediated reactivation of the 21U sensor (RNAe) transgene resulted in RNA expression levels that were very similar to that of 21U sensor(+) in a *pid-2* mutant background (Fig EV1A). We conclude that loss of PID-2 leads to the stochastic loss of the RNAe status of the 21U sensor, implying a role for PID-2 in the inheritance of silencing.

## PID-2 acts together with HRDE-1 to silence Tc1 transposition

Next, we tested whether *pid-2*(*xf23*) impaired silencing of the DNA transposon Tc1. We found that *pid-2*(*xf23*) animals displayed activation of Tc1, but at a frequency that is significantly below that observed in, for instance, a *wago-1; wago-2; wago-3* triple mutant (Fig 1C). We have previously described that Tc1 silencing depends on the combined activity of PRG-1 and HRDE-1 (de Albuquerque *et al*, 2015), so we also tested double mutants between *pid-2* and these two Argonaute proteins. This revealed enhanced activation of Tc1 reactivation in *pid-2;hrde-1* double mutants, compared with both single mutants. Surprisingly, Tc1 activity was undetectable in *pid-2;prg-1* double mutants (Fig 1C). From this experiment, we conclude that PID-2 plays a role in Tc1 silencing and that is does so primarily together with HRDE-1. Loss of PRG-1 from *pid-2* mutants appears to enhance Tc1 silencing. We will address this unexpected result in the discussion.

## PID-2 is essential for silencing by maternally provided 21U RNAs

We have shown before that maternally provided PRG-1 is required to initiate silencing of a 21U sensor transgene: heterozygous offspring of homozygous *prg-1* mutant mothers displayed strong defects in initiating silencing of a 21U sensor transgene (de Albuquerque *et al*, 2014). The same experiment using *pid-2* mutant mothers revealed similar results: a significant fraction of the offspring of *pid-2* mutant mothers could not induce silencing on a 21U sensor that was brought in via the sperm, despite the fact that this offspring carried a wild-type copy of *pid-2* (Fig 1D). This result reveals that PRG-1 and PID-2 likely act during early development and that the absence of this early function cannot be rescued in the adult germline.

We next tested whether maternally provided 21U RNAs could be sufficient to establish target silencing and whether this would require PID-2 as well. To achieve this, we made use of *pid-1* mutants, which lack 21U RNAs (de Albuquerque *et al*, 2014), and crossed *pid-1* mutant males that express the 21U sensor(+) with *pid-1* heterozygous hermaphrodites that did not carry a 21U sensor

transgene. The *pid-1* homozygous mutant offspring from this cross cannot produce 21U RNAs themselves, but do receive maternal PRG-1 and 21U RNAs. In other words, any 21U RNA-driven activity in these embryos stems from the maternal pool alone. We observed that a large fraction (~ 40%) of such animals was able to silence the 21U sensor(+), indicating that the maternal 21U RNAs were sufficient for silencing (Fig 1E). Interestingly, even though these animals were not able to make 21U RNAs, the silencing that had been established was transmitted stably for many generations, suggesting that an RNAe-like state had been induced by just the transient exposure to maternal 21U RNAs.

We sequenced small RNAs from two strains isolated from these crosses: one in which the 21U sensor had become silenced (OFF) and one in which it had remained active (ON). This revealed the absence of 22G RNAs targeting the 21U sensor in the ON strain and a typical RNAe-like 22G RNA pattern in the OFF strain (Fig EV1G). Furthermore, the silencing induced by maternal 21U RNAs was found to be lost in *hrde-1* mutants (Fig EV1H). Finally, maternal 21U RNAs could not induce any silencing in *pid-2(xf23)* mutants (Fig 1F). We conclude that maternal 21U RNAs can be sufficient to induce an RNAe status on a 21U sensor transgene and that this process requires PID-2.

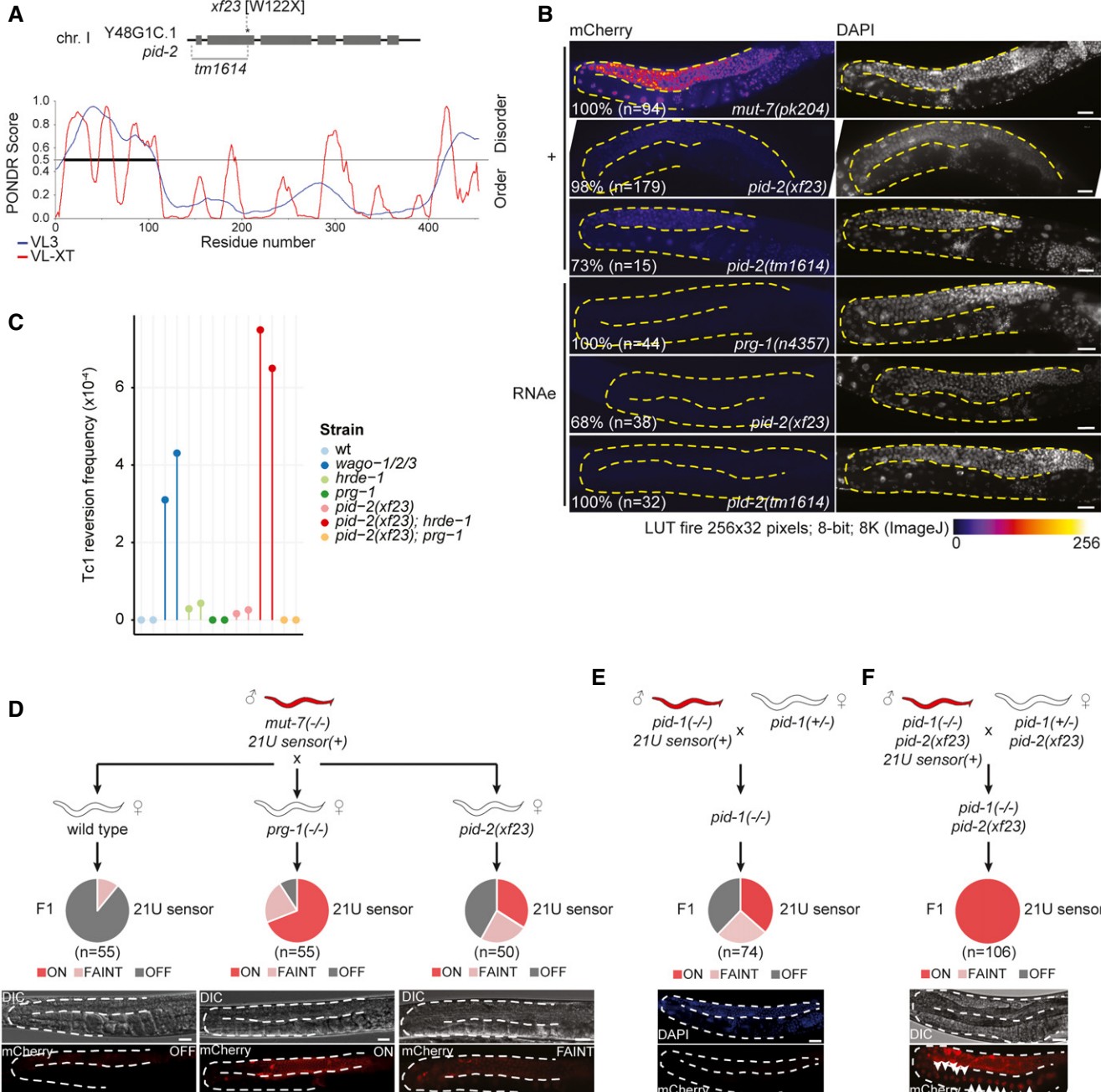

Figure 1.

◀

**Figure 1.  PID-2 is a novel factor required for establishing *de novo* silencing mediated by maternally provided small RNAs.**

A  Schematic representation of the Y48G1C.1/ *pid-2* gene and its mutant alleles (*xf23* and *tm1614*). The line plot displays the predicted disorder of the Y48G1C.1/PID-2 protein, as obtained from PONDR, using the algorithms VL3 and VL-XT.
B  Expression of the 21U sensor (left) and DAPI staining (right) of gonad arms in the indicated genetic backgrounds. Gonads are outlined by a dashed line. The mCherry signal is represented in pseudo-colours [LUT fire (Image)] to reflect differences in the intensity of the signal. Number of animals analysed and with indicated phenotype are given in the panel. Animals not showing the activation of the 21U sensor(+) were still silenced, and animals that did not show the silenced 21U sensor (RNAe) state were expressing weakly, comparable to the 21U sensor(+). Scale bar: 25 μm.
C  Tc1 reversion assay in different genetic backgrounds. All the strains tested carried the *unc-22::Tc1(st136)* allele. Tc1 excision can result in restoration of *unc-22* function, which can be scored visually. Negative control = *unc-22::Tc1(st136)* in a wild-type background; positive control = *wago-1/-2/-3*. Two independent experiments per strain are represented. See Materials and Methods for details.
D  Crossing scheme to address the re-initiation of silencing of the 21U sensor. A *mut-7* mutant male expressing the 21U sensor is crossed with either a wild type (left), *prg-1* (middle) or *pid-2* (right) mutant hermaphrodite. Their F1 offspring was scored for expression of the 21U sensor by microscopy. Three states of expression were scored and represented in a pie chart. The three expression states are exemplified by representative images at the bottom: OFF (left), FAINT (right) or ON (middle). DIC images are shown above the fluorescence panels. Gonads are outlined by a dashed line. Scale bar: 25 μm.
E  Crossing scheme to address whether maternal 21U RNAs are sufficient to re-initiate the silencing of the 21U sensor. A *pid-1* mutant male expressing the 21U sensor is crossed with a hermaphrodite, heterozygous for the same mutation. All their F1 offspring inherit a pool of 21U RNAs from the hermaphrodite, but in 50% of the F1, which is *pid-1* homozygous mutant, no zygotic PID-1 is present, hence no zygotic 21U RNAs can be made. The silencing or expression of the 21U sensor in the *pid-1* homozygous mutant F1 has been scored by microscopy and depicted in a pie chart. At the bottom, a representative image of an animal carrying a silenced 21U sensor (lower: mCherry signal; upper: DAPI staining) in *pid-1* mutant offspring. Gonads are outlined by a dashed line. Scale bar: 25 μm.
F  Crossing scheme to test the role of PID-2 in re-initiating the silencing of the 21U sensor, mediated by maternally provided 21U RNAs only. The expression of the 21U sensor in the F1 has been scored by microscopy and depicted in a pie chart. White arrowheads indicate the many arrayed oocytes, typical of a feminized germline. DIC and fluorescence image of a representative animal are shown at the bottom. Gonads are outlined by a dashed line. Scale bar: 25 μm.

Source data are available online for this figure.

We note that the *pid-2* mutant experiment depicted in Fig 1F revealed an unexpected effect: all F1 from the cross that were *pid-1; pid-2* homozygous mutant, but not those with a wild-type copy of *pid-1*, showed a developmental phenotype. Specifically, these animals were feminized, as evidenced by the characteristically arrayed oocytes lined up before the spermatheca (Figs 1F and EV1I) and the fact that we could rescue their sterility by mating to a wild-type male. This was surprising, as we were able to make and maintain a *pid-1;pid-2* homozygous mutant strain to start this experiment. However, upon careful investigation of this strain, we did observe feminized individuals (3/30 animals). In addition, a pseudo-male animal (1/30 animals) was detected (Fig EV1I). It has been shown that a specific 21U RNA acts in the sex determination pathway, thereby playing a role in proper gonad development (Tang *et al*, 2018). Possibly, this is related to our observation, but it currently remains unclear why the feminization phenotype was so much more prominent in the crosses than in the established double mutant strain.

**Loss of PID-2 causes a reduction in 21U sensor-derived 22G RNAs**

We next performed small RNA sequencing on gravid adults to uncover defects in small RNA populations, which could explain the 21U sensor reactivation in *pid-2* mutants. We sequenced at least triplicates of each strain. First, we checked 21U RNA levels, but found these to be virtually unchanged (Fig EV2A). Hence, the 21U sensor silencing defect was not due to loss of 21U RNAs. We then checked 22G RNAs that are derived from the 21U sensor transgene. As controls, we sequenced wild-type animals carrying a silenced sensor, *prg-1* mutant strains in which the sensor was either expressed or not (RNAe), and *hrde-1* and *mut-7* mutants in which the RNAe status was disrupted (Fig 2). In wild-type animals, two populations of 22G RNAs could be seen: one that is close to the indicated 21U RNA recognition site and one that spreads along the mCherry coding region. The one close to the 21U RNA binding site has been named

secondary 22G RNAs (Sapetschnig *et al*, 2015). They are likely triggered directly by PRG-1, as this population is gone in *prg-1* mutants in which the sensor is active, but are much less affected by loss of HRDE-1 (Fig 2). The pool produced along the mCherry coding sequence has been dubbed tertiary 22G RNAs (Sapetschnig *et al*, 2015) and was found to be dependent on HRDE-1. Loss of MUT-7 strongly affected both secondary and tertiary 22G RNA pools (Fig 2).

We then analysed the effect of *pid-2(xf23)* and *pid-2(tm1614)* on these 22G RNA populations. For this, we crossed the 21U sensor into *pid-2* mutants, either from a *mut-7* mutant background, in which it was expressed (+), or from a *prg-1* mutant background, in which it was under control of RNAe (Fig 2). Both *pid-2* alleles basically produced the same results. First, the secondary 22G RNAs were reduced compared with wild-type and *hrde-1* mutants. This was true whether the transgene originally was under RNAe or not (Figs 2 and EV2B). However, consistently fewer secondary 22G RNAs were detected when the sensor was originally under RNAe (Figs 2 and EV2B). Reduced secondary 22G RNA coverage was also found on endogenous PRG-1 target sites (Fig EV2C and D). These results show that the direct 22G RNA response to PRG-1 is impaired, but not absent in *pid-2* mutants. Second, tertiary 22G RNAs were almost completely lost when the 21U sensor was introduced in an active (+) state (Figs 2 and EV2B). In contrast, tertiary 22G RNAs were reduced, but still clearly detectable when the 21U sensor was introduced in an RNAe state (Figs 2 and EV2B). Altogether, we conclude that lack of PID-2 dampens the overall production of both secondary and tertiary 22G RNA populations on the 21U RNA sensor, but does not eliminate it.

**PID-2 affects endogenous 22G and 26G RNA populations**

We also checked the effect of PID-2 on other classes of endogenous small RNAs. As expected, miRNAs were unaffected in *pid-2* mutants (Fig EV2A). Interestingly, the strongest effect we observed on total pools of small RNA types was on 26G RNAs (Fig EV2A), and 22G

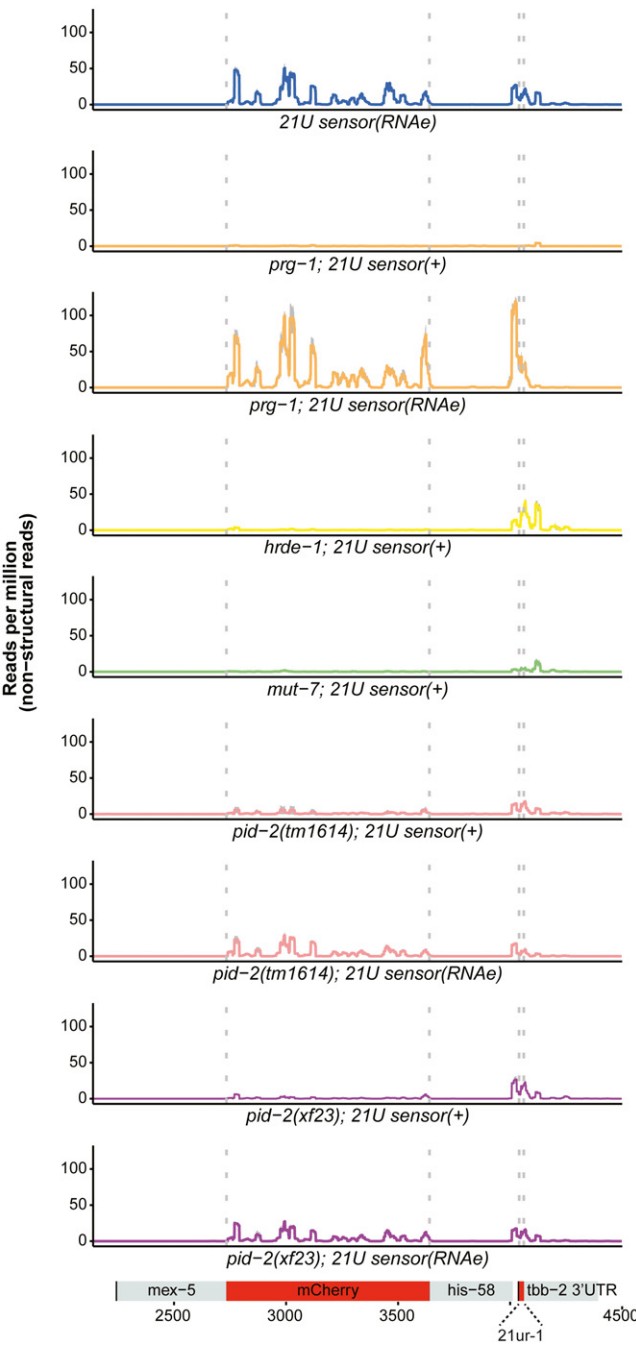

**Figure 2. Small RNA sequencing of 22G RNAs mapping to the 21U sensor in *pid-2* mutants.**

The 22G RNAs mapping to the 21U sensor transgene were identified from small RNA sequencing data, and read density was plotted over the transgene, which is schematically depicted at the bottom. The aggregated results of three replicates are shown for each indicated genotype in the different panels. The shading, in grey, represents the standard deviation of the read density from the three replicates. (+) means that the sensor was detectably expressed. (RNAe) means that the sensor was crossed into the respective mutant background in an RNAe state (i.e. its silencing was PRG-1-independent), and its expression remained undetectable by microscopy.

RNAs produced from ERGO-1 targets were also mildly reduced in *pid-2(xf23)* (Fig EV2C). Consistent with our earlier suggestion that *tm1614* may be a weaker allele of *pid-2*, *pid-2(tm1614)* mutants did not show this effect on either 26G RNAs (Fig EV2A) or associated ERGO-1 22G RNAs (Fig EV2C). The effect on overall bulk 22G RNA levels was only minor (Fig EV2A). Also when we split the 22G RNAs into previously defined sub-categories (Gu *et al*, 2009; Bagijn *et al*, 2012; Conine *et al*, 2013; Phillips *et al*, 2014; Zhou *et al*, 2014; Almeida *et al*, 2019a), we only identified relatively small differences in all pathways analysed (Fig EV2C).

These bulk analyses are blind to potentially strong effects on individual genes. We therefore performed a differential targeting analysis to identify potential individual genes that either gained or lost 22G RNAs in *pid-2* mutants. This revealed that many genes displayed consistently either up- or down-regulated 22G levels in *pid-2(xf23)* mutants (Fig 3A). Specifically, with a cut-off at twofold change and adjusted *P*-value < 0.05, we detected 1,174 genes that lost 22G RNAs and 1,302 genes that gained 22G RNAs (Fig 3A; Table EV1). We asked whether these two sets of genes overlapped significantly with gene sets defined previously as Mutator, CSR-1, ALG-3/-4 or ERGO-1 targets (Fig 3B), and found that the PID-2-responsive genes were enriched for Mutator targets. For *pid-2(tm1614)*, very similar effects were found (Appendix Fig S1A and B; Table EV1), and the up- and down-regulated genes detected in both alleles overlapped strongly (Appendix Fig S1C). We can currently not explain why some genes gained and other genes lost 22G RNAs, but given that in both *pid-2* alleles the same genes gained or lost 22G RNAs, it appears to be a specific effect. This dual effect is strikingly similar to what has been described for *znfx-1* mutants (Ishidate *et al*, 2018). Loss of ZNFX-1 additionally revealed a remarkable change in 22G RNA distribution over the length of the gene body of target loci: 22G RNAs were mostly lost from their 3′ end, whereas 22G RNAs from the 5′ part of these genes increased. This effect was strongest on Mutator targets, but also detectable on CSR-1 targets (Ishidate *et al*, 2018). We therefore probed 22G RNA coverage on the gene bodies of Mutator and CSR-1 targets, using a metagene analysis as employed by Ishidate *et al* (2018), splitting the targets into those that lost or gained 22G RNAs in the two *pid-2* alleles. This revealed that both Mutator and CSR-1 targets that lost 22G RNAs lost them all over the gene body, whereas those that gained 22G RNAs still lost 22G RNAs from their 5′ ends (Fig 3C and D, Appendix Fig S2A–F). We conclude that PID-2 affects 22G RNA production from many loci, including many previously defined Mutator targets, and that these can either gain or lose 22G RNAs. In addition, PID-2 most strongly affects 22G RNA production from the 5′ parts of transcripts, suggesting a potential role in RdRP processivity. We note that the latter effect is opposite to that of ZNFX-1 (Ishidate *et al*, 2018).

## PID-2 interacts with two eTudor domain proteins: PID-4 and PID-5

To define the molecular environment of PID-2, we performed immunoprecipitation (IP) followed by quantitative label-free mass spectrometry (MS) on gravid adults, using both the transgenic line expressing the rescuing, C-terminally tagged PID-2::eGFP fusion protein (*xfSi145*), and a polyclonal antibody that we raised against

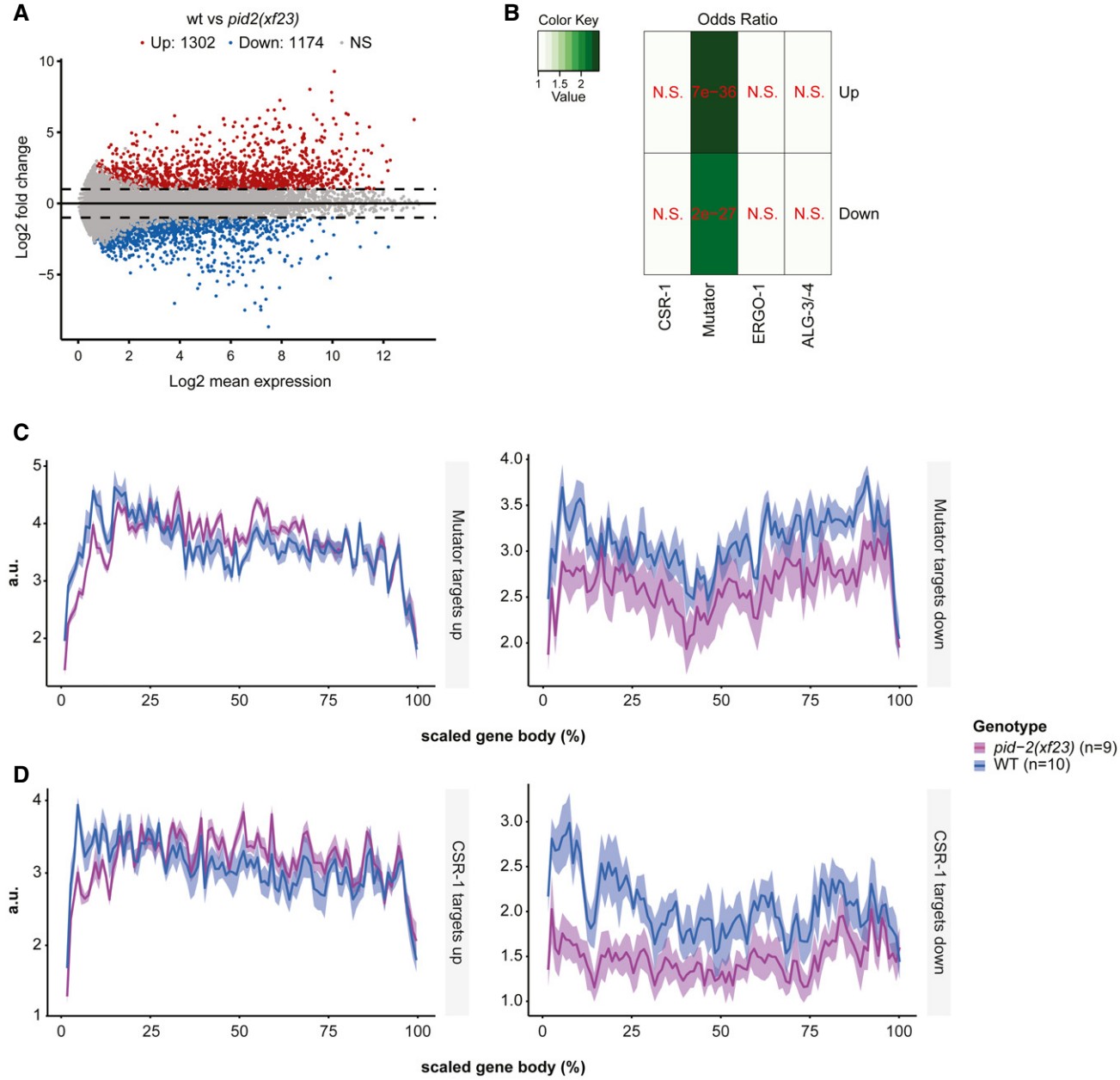

**Figure 3. Loss of PID-2 causes a disbalance in 22G RNA populations.**

A    MA plot of log$_2$ fold changes (*y*-axis) versus the mean of normalized counts of 22G RNAs (*x*-axis) for *pid-2(xf23)* mutants, compared with wild type. Red dots: genes with adjusted *P*-value < 0.05 and fold change > 2. Blue dots: genes with adjusted *P*-value < 0.05 and fold change < −2.

B    Heat map displaying overlap significance between different gene sets and genes that either up- or down-regulated in *pid-2(xf23)* mutants. Significance was tested with Fisher's exact test and *P*-values adjusted with the Benjamini–Hochberg procedure. Colour scheme represents the odds ratio of overlaps representing the strength of association. N.S.: not significant.

C, D  Cumulative 22G coverage along the gene body of Mutator and CSR-1 targets. Values represent 22G coverage normalized to the total coverage of each gene, for wild type (N2) and *pid-2(xf23)* mutants. Gene sets are previously defined 22G RNA target sub-types (see Appendix). The lines represent the average of biological replicates, whereas the shading represents the standard deviation of biological replicates. a.u.: arbitrary units.

the endogenous protein (Fig 4A and B). In addition, we also performed this experiment with an N-terminally 3xFLAG-tagged PID-2 fusion protein expressed from two independent, randomly inserted transgenes generated using the miniMos procedure (Frøkjær-Jensen *et al*, 2014; Fig EV3A and B). In all cases, we could

reproducibly pull down PID-2, indicating that the IP was working well. In addition to PID-2, we consistently identified two non-characterized proteins: W03G9.2 and Y45G5AM.2, even if enrichment of the latter did not reach our significance cut-off in the PID-2::eGFP IP. We named these proteins PID-4 and PID-5, respectively (Figs 4A

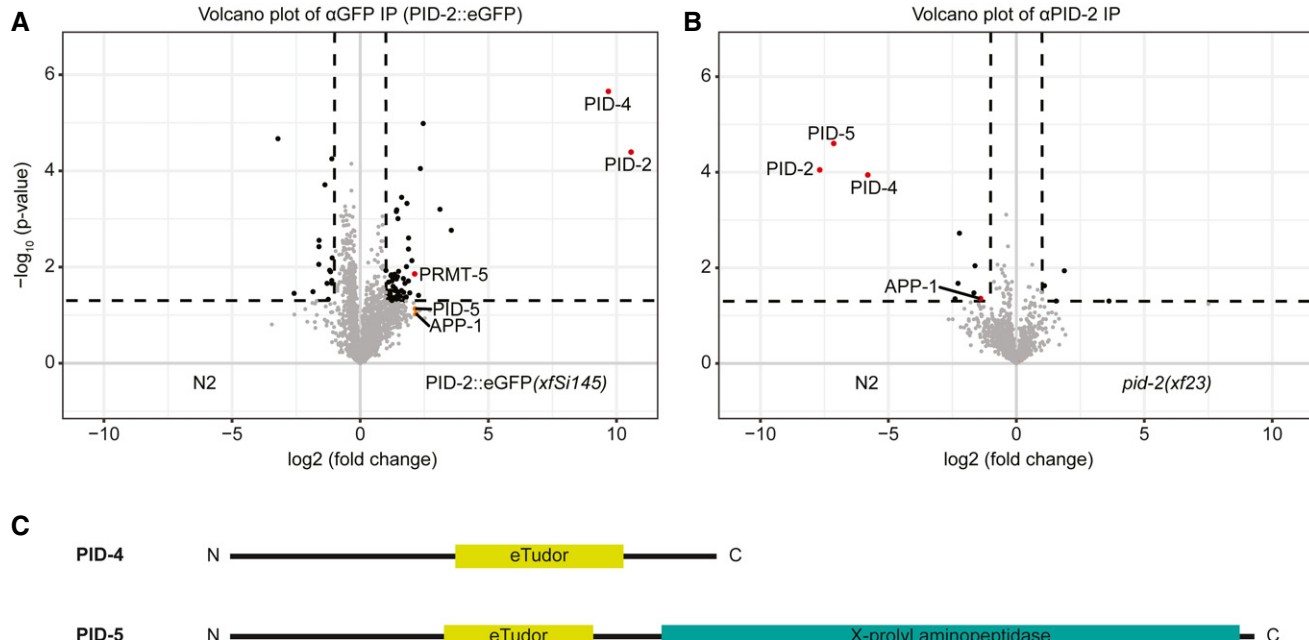

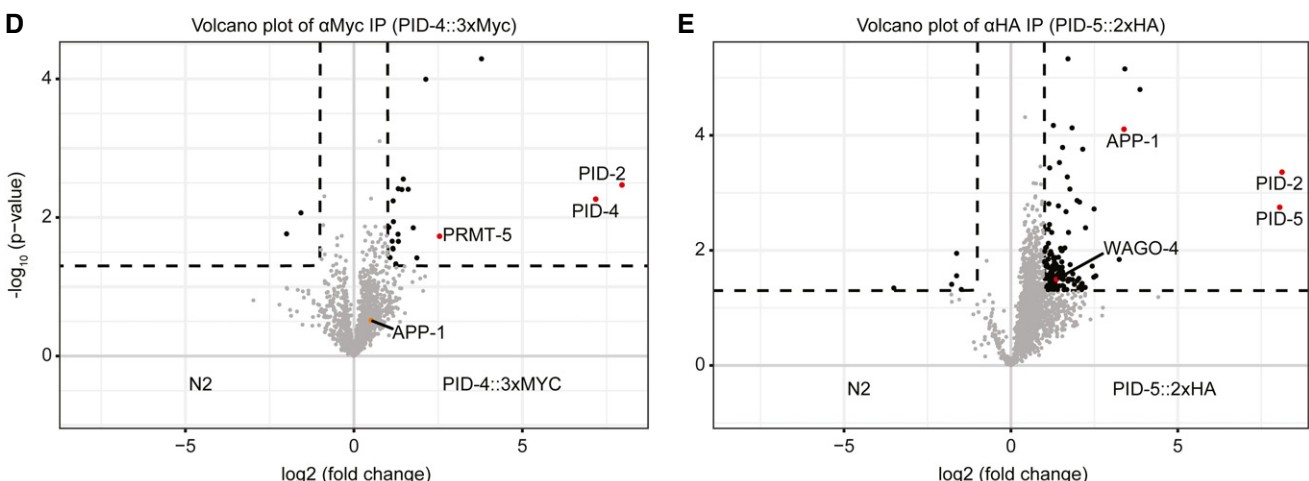

**Figure 4. IP-MS on PID-2 identifies two novel interacting proteins, PID-4 and PID-5, and reveals the existence of two distinct complexes.**

A  Volcano plot representing the enrichment of proteins interacting with PID-2, isolated by immunoprecipitation of eGFP::PID-2 protein, followed by quantitative label-free mass spectrometry. As control, eGFP IPs were performed on protein extracts from wild-type, non-transgenic animals. For each IP-MS experiment, quadruplicates were measured and analysed. The dashed line reflects a significance threshold of *P*-value < 0.05 at twofold enrichment.

B  Volcano plot, as described in (A). In this experiment, the endogenous PID-2 protein was immunoprecipitated, and *pid-2(xf23)* mutant protein extracts were used as control.

C  Schematic representation of the predicted domain structure of PID-4 and PID-5. The eTudor domains were identified using HHpred (Zimmermann *et al*, 2018). The X-prolyl aminopeptidase domain was identified using BLAST (https://blast.ncbi.nlm.nih.gov/Blast.cgi).

D  Volcano plot, as described in (A), representing the enrichment of proteins interacting with PID-4, by immunoprecipitating endogenously tagged PID-4 protein. IPs from protein extracts from wild-type, non-tagged animals were used as control.

E  Volcano plot, as described in (A), representing the enrichment of proteins interacting with PID-5, by immunoprecipitating endogenously tagged PID-5 protein. IPs from protein extracts from wild-type, non-tagged animals were used as control.

and B, and EV3A and B). Both PID-4 and PID-5 are predicted to have an extended Tudor (eTudor) domain, as found by HHpred (Zimmermann *et al*, 2018; Fig 4C). These domains are known to bind symmetrically di-methylated arginines, involving a set of four

conserved aromatic residues that form a so-called aromatic cage and a characteristic acidic amino acid (Gan *et al*, 2019). Neither the aromatic cage nor the acidic residue are found in the PID-4 and PID-5 eTudor domains (Fig EV3C), suggesting that PID-4 and PID-5 do

not bind di-methylated arginines. Interestingly, PRMT-5, the enzyme responsible for symmetric dimethylation of arginine, also displayed enrichment in PID-2::eGFP IPs (Fig 4A). In addition to an eTudor domain, PID-5 also has an X-prolyl aminopeptidase domain, which is very similar to APP-1 (Figs 4C and EV3D). APP-1 is a strongly conserved enzyme, found from yeast to human, that cleaves the most N-terminal amino acid from a polypeptide, provided that the second amino acid is a proline (Laurent *et al*, 2001; Iyer *et al*, 2015). The PID-5 X-prolyl aminopeptidase domain is likely catalytically inactive, as the residues required to coordinate the two $Zn^{2+}$ ions are not conserved (Fig EV3D). Interestingly, in the PID-2 IP-MS experiments APP-1 tended to be enriched (Figs 4A and B, and EV3A and B), even though its enrichment did not always reach our stringent significance cut-off. Since APP-1 itself dimerizes (Iyer *et al*, 2015), this could reflect the presence of PID-5:APP-1 heterodimers (also see below).

We tagged both PID-4 and PID-5 endogenously with an epitope and with a fluorescent tag (Fig EV3E and F), in order to perform IP-MS experiments and to investigate their expression. IP-MS on both PID-4 (Fig 4D) and PID-5 (Fig 4E) enriched for PID-2, consistent with their enrichment in PID-2 IPs. In addition, PRMT-5 and APP-1 were clearly enriched in PID-4 and PID-5 IPs, respectively (Fig 4D and E), lending support to the detection of APP-1 and PRMT-5 in the above-mentioned PID-2 IPs. We did not retrieve PID-5 in PID-4 IPs, or *vice versa*, indicating that PID-4 and PID-5 do not simultaneously interact with PID-2. Consistent with this finding, IP-MS on PID-2 in *pid-4* and *pid-5* mutant backgrounds (Fig EV3E and F) still retrieved PID-5 and PID-4, respectively (Fig EV3G and H). Finally, in relation to the below described relation to Z granules, we note that we identified WAGO-4 in PID-5 IPs (Fig 4E). Collectively, these data identify PID-4 and PID-5 as robust PID-2-interacting proteins. Additionally, APP-1 may interact with the PID-2 complex via PID-5, and PRMT-5 via PID-4. All significantly enriched proteins of the described IPs are provided in Table EV2.

## PID-4 and PID-5 are partially redundant

We generated deletion alleles of *pid-4* and *pid-5* (Fig EV3E and F) and found that these did not show obvious developmental defects. Given that *pid-2* mutants have defects in the silencing of the 21U sensor, we next investigated the expression of the 21U sensor (RNAe or (+)) in *pid-4* and *pid-5* mutants. Independent of the initial status of the 21U sensor, both *pid-4* and *pid-5* mutants were silencing-proficient (Fig EV1A). We hypothesized that PID-4 and PID-5 could be redundant, given that their respective eTudor domains are very similar and could have interchangeable roles (Fig EV3C). A *pid-4;pid-5* double mutant strain indeed revealed redundancy, as these double mutants fail to fully silence the 21U sensor(+) (Fig 5A), with expression levels that are very similar to those found in *pid-2* mutants (Fig EV1A). The 21U sensor (RNAe) was initially not reactivated, as analysed by qRT–PCR (Fig EV1A), but like in *pid-2* mutants, loss of RNAe status and gain of detectable expression could be detected after prolonged culturing (Fig EV1F). We also performed small RNA sequencing on *pid-4*, *pid-5* and *pid-4;pid-5* mutants, to assess effects on 21U sensor-derived 22G RNAs. Both single mutants did not affect the sensor-derived 22G RNAs (Appendix Fig S3A). However, in *pid-4;pid-5* double mutants secondary and tertiary 22G RNA reads from a 21U

sensor (RNAe) dropped significantly, comparable to what we observed in *pid-2* mutants (Figs 5B and EV4A). These results show that PID-4 and PID-5 act redundantly with regard to 21U sensor silencing.

We also assessed the effects of PID-4 and PID-5 on endogenous small RNAs. We could not detect striking alterations in the bulk abundance of any of the small RNA classes in *pid-4*, *pid-5* and *pid-4; pid-5* mutants, although 26G RNAs tended to be reduced (Appendix Fig S3B). When 22G RNA targets were split up into functional sub-categories, *pid-4;pid-5* double mutants displayed a modest, but highly significant loss of 22G RNAs from Mutator and PRG-1 targets, similar to *pid-2* mutants (Appendix Fig S3C). Finally, we analysed 22G RNA abundance per gene in *pid-4*, *pid-5* and double mutants. Loss of PID-4, like loss of PID-2, resulted in genes losing or gaining 22G RNAs (Fig 5C). However, fewer genes were affected in *pid-4* mutants compared with *pid-2* (Fig 5C; Table EV1). The genes that lost 22G RNAs in *pid-4* mutants were strongly enriched for Mutator targets (Fig 5D); the genes that gained 22G RNAs did not show enrichment for any particular gene set we analysed (Fig 5D). Loss of PID-5 showed a strongly asymmetric effect: 161 genes lost, while 1,140 genes gained 22G RNAs (Fig 5E; Table EV1). Both gene sets were enriched for Mutator targets (Fig 5F). The *pid-4;pid-5* double mutant showed gain and loss of 22G RNAs, similar to what we observed for *pid-2* mutants (Fig 5G and H). Comparing the genes that lost and gained 22G RNAs in the various mutants revealed striking overlaps, both in terms of gene identity, and in terms of direction of the detected effect, with the *pid-4;pid-5* double mutant most closely resembling *pid-2* (Appendix Fig S1C). Finally, *pid-4;pid-5* double mutants, but not the respective single mutants, recapitulated the loss of 22G RNAs from the 5′ ends of targets that overall gained 22G RNAs that we observed in *pid-2* mutants (Fig EV4B and C; Appendix Fig S4A–D), supporting the significance and specificity of this effect. These data show that PID-4 and PID-5 have partially redundant roles in regulating 22G RNA production from Mutator target genes and that PID-2 is required for their function.

## PID-2, PID-4 and PID-5 are required for an immortal germline

As mentioned earlier, *prg-1* mutants display a gradual decline of fertility over successive generations. This so-called mortal germline phenotype (Mrt) is not only a characteristic of *prg-1* mutants (Simon *et al*, 2014), but also of mutants for other factors participating in the RNAe machinery, such as *hrde-1*, *nrde-1/-2/-4* (Buckley *et al*, 2012), the H3K4 methyltransferase *set-2* (Xiao *et al*, 2011), the H3K9 methyltransferase homolog *set-32* (Spracklin *et al*, 2017) and two factors involved in RNAi inheritance, WAGO-4 (Xu *et al*, 2018) and ZNFX-1 (Wan *et al*, 2018). Therefore, we tested whether *pid-2*, *pid-4*, *pid-5* or *pid-4;pid-5* mutants showed a Mrt phenotype. As expected, both *prg-1* and *hrde-1* mutants started to show fertility defects already after few generations and eventually became sterile between 6 and 14 generations, whereas the large majority of wild-type worms did not become sterile, even after 47 generations. The different Pid mutants, including the *pid-4* and *pid-5* single mutants, also showed a Mrt phenotype, even though the onset and progression were slower than in the *prg-1* and *hrde-1* mutants (Fig 5I and J). We note that the presented data tend to under-estimate the effect of the mutations, as we noticed that the numbers of offspring

produced by the various mutants already dropped after a few generations, and this is not reflected in the data.

Accumulation of 22G RNAs that target replicative histone mRNAs has been identified as a cause of the Mrt phenotype in mutants lacking 21U RNAs (Barucci *et al*, 2020). This prompted us to look at such 22G RNA populations in our mutants. However, we did not detect a gain for the either of the four replicative histone classes; rather, they tended to show mild depletion in the various mutants (Appendix Fig S5A and B). While this result may indicate a different basis for the Mrt phenotype, we note that our experimental set-up for small RNA sequencing did not specifically address late generations that are close to sterility. We conclude that PID-2, PID-4 and PID-5 are required for germline immortality through an as yet undefined mechanism.

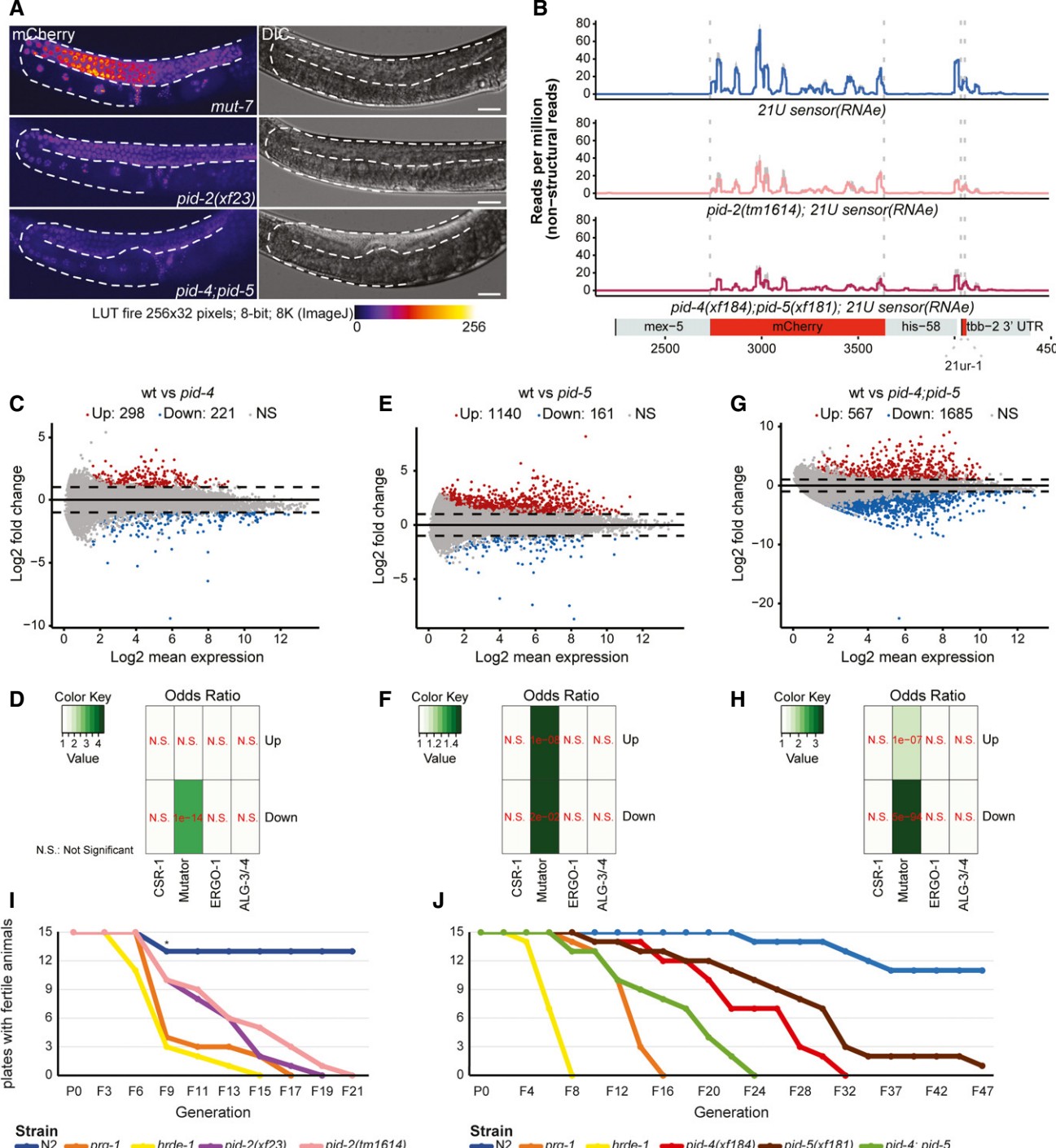

**Figure 5.**

◀

**Figure 5.  PID-2 and its interactors, PID-4 and PID-5, are required for maintenance of an immortal germline.**

A    Expression of the 21U sensor in the indicated genetic backgrounds. Gonads are outlined by a dashed line. The mCherry signal is represented in pseudo-colours [LUT fire (ImageJ)] to reflect differences in the intensity of the signal. The panels on the right are DIC images of the respective animals. Scale bar: 25 μm.

B    22G RNA coverage of the 21U sensor (RNAe) in the indicated genetic backgrounds. Quantification of the secondary and tertiary 22G RNAs is provided in Fig EV4A.

C    MA plot of $\log_2$ fold changes (on the *y*-axis) versus the mean of normalized counts of 22G RNAs (on the *x*-axis) for *pid-4* mutants, compared with wild type. Red dots: genes with adjusted *P*-value < 0.05 and fold change > 2. Blue dots: genes with adjusted *P*-value < 0.05 and fold change < −2.

D    Heat map displaying overlap significance between different gene sets and genes that either up- or down-regulated in *pid-4* mutants. Significance was tested with Fisher's exact test and *P*-values corrected with the Benjamini–Hochberg procedure. Colour scheme represents the odds ratio of overlaps representing the strength of association.

E    As panel (C), but for *pid-5* mutant animals.

F    As panel (D), but for *pid-5* mutant animals.

G    As panel (C), but for *pid-4;pid-5* double mutant animals.

H    As panel (D), but for *pid-4;pid-5* double mutant animals.

I, J  Line plots representing fertility over generations for the indicated genetic backgrounds at 25°C. Six L2-L3 larvae for each of the indicated backgrounds were hand-picked to a fresh plate every 4 or 6 days, counting as two or three generations, respectively, until no larvae were present on the plate to be picked. *: two plates of N2 were contaminated and were excluded from the assay.

Source data are available online for this figure.

## PID-2, PID-4 and PID-5 localize to perinuclear granules in germ cells

We performed confocal microscopy to investigate the expression pattern and localization of PID-2, tagged with eGFP at its C-terminus (*xfSi145*; Fig EV1B). We imaged L4 larvae, as the expression levels of PID-2, as well as of PID-4 and PID-5 (see below), were rather low and most clearly detected in the pachytene stage of the meiotic region, which is most extended at the L4 stage. PID-2::eGFP localized to perinuclear granules. As shown by the colocalization with PGL-1 (Kawasaki *et al*, 1998), PID-2 foci were adjacent to P granules (Fig 6A). Z granules, marked by ZNFX-1, have been recently described to be juxtaposed to P granules and involved in transmitting genetic information to the next generation via the oocytes (Ishidate *et al*, 2018; Wan *et al*, 2018). As we observed a role for PID-2 in RNAe inheritance, PID-2 may well localize to Z granules. The typical distance between PID-2 and PGL-1 foci (Fig 6F), and ZNFX-1 and PGL-1 foci (see below and Fig EV5A) closely matched each other, consistent with this idea. Unfortunately, the direct assessment of colocalization between ZNFX-1 and PID-2 was thus far hampered by the very close linkage between the PID-2::eGFP transgene and an available endogenously tagged ZNFX-1 allele (Wan *et al*, 2018). Given that Wan *et al* (2020) show that PID-2 (named ZSP-1 by Wan et al.) is indeed in Z granules, we did not further pursue PID-2-ZNFX-1 colocalization ourselves.

PID-4 and PID-5 were both endogenously tagged at the C-terminus with mTagRFP-T (Fig EV3E and F). We found that PID-4 and PID-5 were also specifically expressed in germ cells and localized to perinuclear granules as well. PID-5 was detectable in foci around relatively few nuclei at the pachytene stage, whereas PID-4 was found throughout the gonad, with stronger expression at the pachytene stage (Fig 6B–E, G and H). PID-4 and PID-5 colocalized to a large extent with the P granule marker DEPS-1 (Spike *et al*, 2008; Fig 6B and C), and the typical distance between PID-4/-5 foci and DEPS-1 was significantly shorter than between PID-2 and PGL-1 (Fig 6F). These data suggest PID-4 and PID-5 may be in P granules. However, PID-4 and PID-5 were closer to ZNFX-1 (Fig 6D and E), or PID-2 (Fig 6G and H) foci than P granules are (Figs 6F and EV5A), raising the possibility that PID-4 and PID-5 foci are distinct from P granules. Finally, we checked whether PID-4 and PID-5 may be in

SIMR-1 foci, a recently described germ granule distinct from P and Z granules (Manage *et al*, 2020), even though SIMR-1 was not detected in our IP-MS analyses. In general, SIMR-1 foci were fewer in number, and were also more restricted to the mitotic zone, whereas PID-4 and PID-5 foci are found more in the pachytene zone (Fig EV5B). When present on the same nuclei, the foci were often positioned close to each other, but at distances similar to what we measure for PGL-1 and ZNFX-1 (Appendix Fig S6A).

We conclude that PID-2 on the one hand, and PID-4 and PID-5 on the other, displays distinct subcellular localization in discrete perinuclear foci. While PID-2 is in Z granules, PID-4 and PID-5 are found in different granules, very close to and partially overlapping with P granules.

## PID-2, PID-4 and PID-5 affect Z granule formation

Finally, we checked whether loss of PID-2, PID-4 or PID-5 affected P or Z granules. Localization of PGL-1 strongly resembled that of wild-type animals in all mutants tested, including *pid-4;pid-5* double mutants (Fig 7A–E). In contrast, in *pid-2* and in *pid-4;pid-5* double mutants Z granules were affected (Fig 7B and C). First, we noticed the appearance of relatively large Z granules in *pid-2* mutants. To quantify this, we measured the surface of Z granules and compared these to the area of Z granules in the various wild-type strain. Even though many Z granules were similarly sized in the various genetic backgrounds, Z granules indeed displayed a tendency to be larger in *pid-2* mutants than in wild-type animals (Fig EV5C). This effect was not seen in *pid-4*, *pid-5* and *pid-4;pid-5* double mutants (Fig EV5C). Second, the number of Z granules appeared to be lower. Hence, we counted the number of P and Z granules, to determine their ratio. This revealed a significant loss of Z granules compared with P granules in *pid-2* and in *pid-4;pid-5* double mutants (Fig 7F). Z granules did remain distinct from P granules (Figs 7A–E, and EV5A). In fact, the distance between Z and P granules tended to be slightly longer in *pid-2* mutants (Fig EV5A), but this could be a result of the tendency of Z granules to be larger (Fig EV5C). We also checked PID-4 and PID-5 localization in *pid-2* mutants. This revealed that PID-4 foci were affected in *pid-2* mutants. PID-4 foci can still be observed, but they were fewer than in wild-type animals and the intensity of the signal was lower, indicating that PID-4 expression

was reduced in the absence of PID-2, and/or that PID-4 localization was affected (Fig 7G, Appendix Fig S6B). PID-5 foci were not affected by loss of PID-2 (Fig 7H, Appendix Fig S6B). Like P granules (Fig EV5C), the remaining PID-4 foci in *pid-2* mutants were further away from ZNFX-1 compared with wild type (Fig 7I).

Interestingly, this increase in distance was not detected for PID-5 foci (Fig 7I), suggesting that this effect is not due to the increase in Z granule size. To check whether the observed effects on Z granules could stem from effects on ZNFX-1 stability, we performed Western blot analysis on endogenously tagged ZNFX-1 in the different

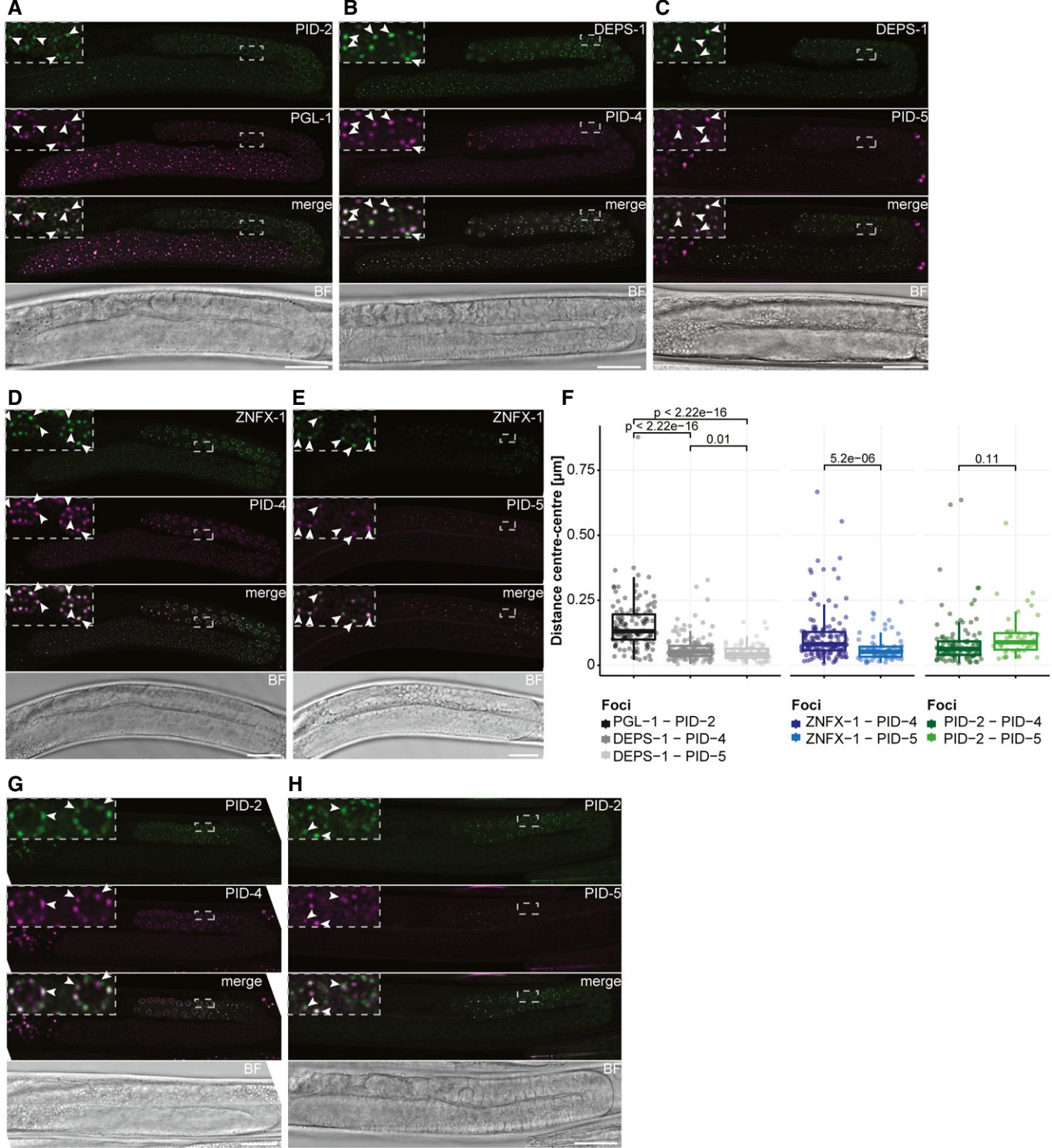

**Figure 6.**

**Figure 6.  PID-2, PID-4 and PID-5 are present in different germ granules.**

A–C      Expression of PID-2::eGFP(*xfSi145*) (A), PID-4::mTagRFP-T (B) and PID-5::mTagRFP-T (C) in perinuclear granules in the germline. PGL-1::mTagRFP-T (A) and DEPS-1::
         GFP (B, C) mark P granules. The indicated dashed boxes reflect zoom-ins on specific nuclei to better visualize the granules, and their overlaps. One L4 gonad is
         shown for each animal. Note that most of the L4 gonad is in pachytene stage. Arrowheads indicate individual condensates. Scale bar: 25 μm.

D, E     Expression of PID-4::mTagRFP-T (D) and PID-5::mTagRFP-T (E) together with 3xFLAG::GFP::ZNFX-1, a Z granule marker. The indicated dashed boxes reflect zoom-ins
         on specific nuclei to better visualize the granules, and their overlaps. One L4 gonad is shown for each animal. Note that most of the L4 gonad is in pachytene
         stage. Arrowheads indicate individual condensates. Scale bar: 25 μm.

F        Box plots representing the distance (μm) between the centres of two fluorescent signals from the indicated fusion proteins, as represented in panels (A–E, G, H).
         The distance between each pair of fluorescent proteins is represented by a dot. Between 4 and 10 different gonads were analysed for each condition. The median
         is represented by a line. The interquartile range (IQR), 25th–75th percentile, is represented by the upper and lower lines, respectively, and whiskers represent the
         first quartile (down to −1.5*IQR) or the third quartile (up to +1.5*IQR). *P*-values were calculated using an unpaired *t*-test (two-tailed).

G, H     Expression of PID-2::eGFP (*xfSi145*) together with either PID-4::mTagRFP-T (G) or PID-5::mTagRFP-T (H). The indicated dashed boxes reflect zoom-ins on specific
         nuclei to better visualize the granules, and their overlaps. One L4 gonad is shown for each animal. Note that most of the L4 gonad is in pachytene stage.
         Arrowheads indicate individual condensates. Scale bar = 25 μm.

Source data are available online for this figure.

mutant backgrounds (Appendix Fig S6C). This did not reveal major changes in ZNFX-1 levels. Everything considered, we conclude that PID-2, PID-4 and PID-5 affect steady-state Z granule size and number, while not affecting P granules visibly. Additionally, loss of PID-2 versus loss of PID-4/-5 does not perfectly phenocopy each other at this level, indicating that some PID-2 function remains in *pid-4;pid-5* double mutants, and/or that some PID-4/-5 function remains in *pid-2* mutants.

# Discussion

We describe the identification and characterization of three novel proteins, PID-2, PID-4 and PID-5, that are required to establish and maintain silencing on a 21U RNA target, and for normal 22G RNA homeostasis. They localize to distinct germ cell-specific perinuclear granules and affect the morphology of Z granules. We will discuss below potential modes of action of these proteins and in addition touch upon some evolutionary aspects.

## PID-2 is required for *de novo* silencing by 21U RNAs and for inheritance of silencing via 22G RNAs

We have shown that maternally provided 21U RNAs are required and sufficient for *de novo* target silencing by RNAe and that they absolutely require PID-2 to do so. This result identifies embryogenesis as an important developmental period for establishment of 21U RNA-mediated silencing and implies that RNAe is less effectively established in the adult germline. The idea that maternal Piwi activity is particularly important in the PGCs is not unique to *C. elegans*. Also in *Drosophila*, maternal impact of Piwi proteins and piRNAs has been described (Siddiqui *et al*, 2012; Akkouche *et al*, 2017), and in zebrafish, we found significant maternal effects on piRNA populations that differ among strains (Kaaij *et al*, 2013). Additionally, in plants, strong maternal effects of small RNAs have been described, and in unicellular eukaryotes, such as *Tetrahymena* and *Paramecium*, similar mechanisms operate in which parental nuclei produce small RNAs that act in the nuclei generated during mating (Malone & Hannon, 2009; Van Wolfswinkel & Ketting, 2010; Castel & Martienssen, 2013). Hence, the concept that small RNAs from the parents prime effects in the offspring appears to be a broadly conserved aspect of these pathways. It may therefore not be surprising that the PGCs in animal embryos may contain specialized small RNA-related mechanisms compared with the adult germline, and a better understanding of such developmentally regulated aspects will be needed to understand small RNA function.

We also show that PID-2 is required for stable inheritance of RNAe, and Wan *et al* (2020) show that PID-2/ZSP-1 is required for inheritance of RNAi. While at first glance this may hint at a second function for PID-2, next to its role in initiation, we propose that both the initiation and maintenance defects of *pid-2* mutants may stem from one and the same activity: (re-)initiation of silencing within the PGCs of the next generation. In case of *de novo* silencing, maternal 21U RNAs initiate the 22G RNA response, and PID-2 is required to do so effectively. In case of inheritance, inherited 22G RNAs re-trigger 22G RNA production, and again, PID-2 is required for effectivity. The fact that we see a stronger requirement for PID-2 in establishment of silencing compared with maintenance could be due, for instance, to differences in 21U and 22G RNA inheritance. Seen from this perspective, PID-2 may play an important role in interpreting inherited small RNA populations from the parents and in using these to establish silencing within the PGCs in the embryo.

## What is the molecular mechanism behind the observed phenotypes?

We identified PID-2 because it has a role in the 21U RNA pathway, as shown by the defects in silencing of the 21U RNA sensor. Since 21U RNAs are not affected, but 22G RNAs that target the 21U sensor are, we consider a potential function of PID-2 between PRG-1 target recognition and RdRP activity. In addition, PID-2 helps to maintain HRDE-1 mediated silencing. In analogy to PRG-1, we envision a role between HRDE-1 activity and RdRP activity. PID-2 is not essential for silencing, since significant silencing and 22G RNA production can be achieved in *pid-2* mutants. Hence, it seems more likely that PID-2 regulates factors that in turn execute the silencing response. We also note that PID-2 is not specific to the PRG-1 pathway, as *pid-2* mutants do show effects on overall 22G RNAs, including those of the CSR-1 pathway, as well as relatively strong effects on 26G RNAs. Possibly, PID-2 may have a general function in the regulation of RdRP activity and thus affect both 22G and 26G RNAs. The fact that, in *pid-2* mutants, Mutator target genes lose 22G RNAs preferentially from their 5′ regions also supports this idea: PID-2 could be involved in the processivity of the RdRP enzyme RRF-1. We note that this

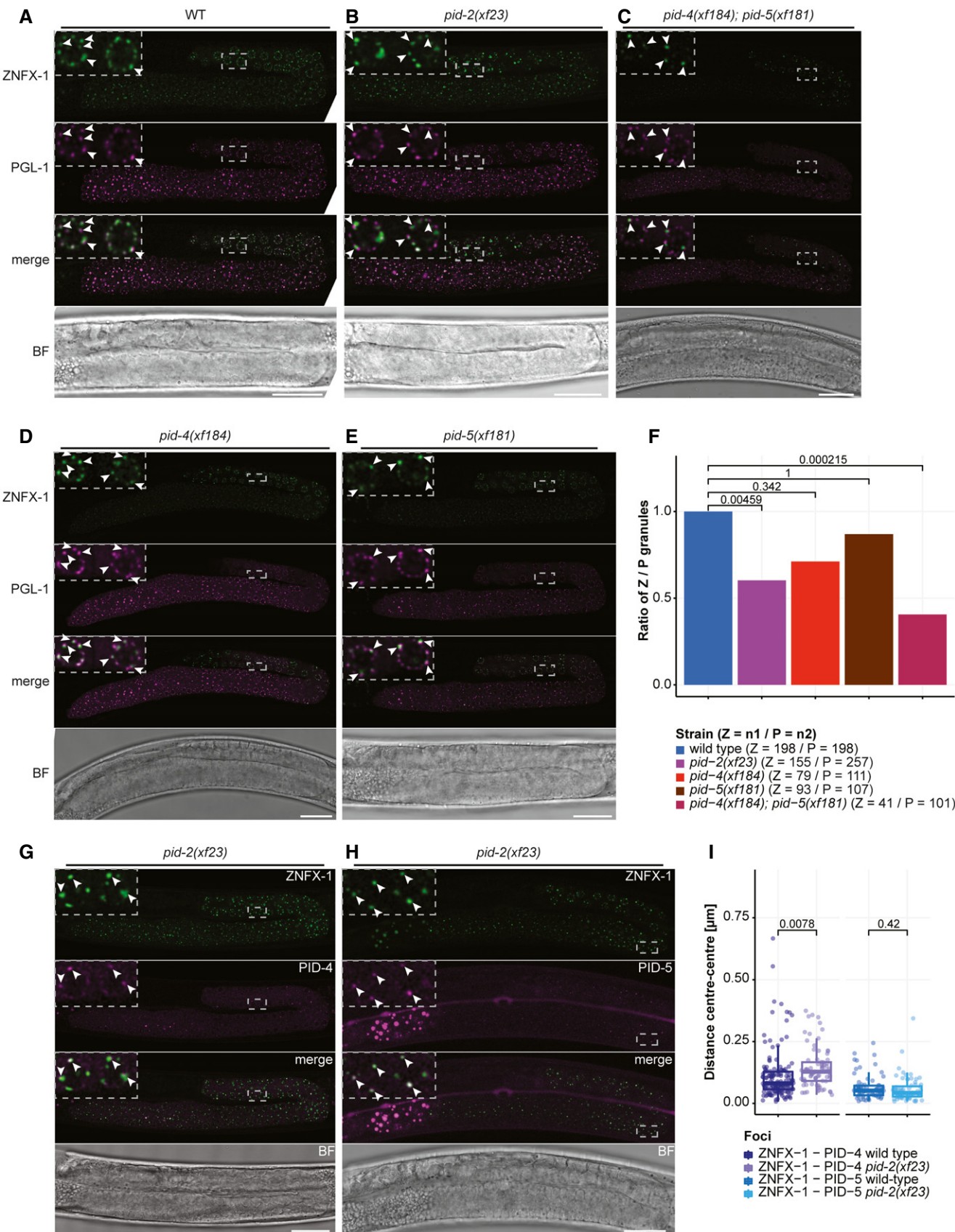

**Figure 7.**

**Figure 7. Z granules are affected by PID-2, PID-4 and PID-5.**

A–E   Expression of 3xFLAG::GFP::ZNFX-1 and PGL-1::mTagRFP-T in a wild type (A), *pid-2* (B), *pid-4;pid-5* (C), *pid-4* (D) and *pid-5* (E) mutant backgrounds. The indicated dashed boxes reflect zoom-ins on specific nuclei to better visualize the granules, and their overlaps. One L4 gonad is shown. Arrowheads indicate individual condensates. Scale bar: 25 μm.

F   Quantification of the ratio Z/P granules in wild type, *pid-2*, *pid-4/-5*, *pid-4* and *pid-5* mutant backgrounds. The number of granules is indicated in brackets, next to the genotype. *P*-values were calculated using a two-proportion z-test and adjusted for multiple comparisons with the Holm method.

G, H   Expression of PID-4::mTagRFP-T (G) and PID-5::mTagRFP-T (H) together with 3xFLAG::GFP::ZNFX-1, a Z granule marker, in a *pid-2* mutant background. The indicated dashed boxes reflect zoom-ins on specific nuclei to better visualize the granules, and their overlaps. One L4 gonad is shown for each animal. Scale bar: 25 μm.

I   Box plots representing the distance (μm) between the centres of two fluorescent signals from the indicated fusion proteins, as represented in panels (G, H) and in Fig 6D and E. Note that the wild-type distance measurements are the same represented in Fig 6F. The distance between each pair of fluorescent proteins is represented by a dot. Between 4 and 10 different gonads were analysed for each condition. The median is represented by a line. The interquartile range (IQR), $25^{th}$–$75^{th}$ percentile, is represented by the upper and lower lines, respectively, and whiskers represent the first quartile (down to −1.5*IQR) or the third quartile (up to + 1.5*IQR). *P*-values were calculated using a *t*-test (two-tailed).

Source data are available online for this figure.

effect is opposite to what has been observed for *znfx-1* mutants that lose 22G RNAs preferentially within the 3′ end of target genes (Ishidate *et al*, 2018). In biological processes, stable states are typically achieved by applying opposite forces; ZNFX-1 and PID-2 may represent two such oppositely acting mechanisms to ensure stable RdRP activity.

However, there are also other aspects that need to be considered. With the available data, it is very difficult to dissect primary from secondary effects. Hence, the effect of PID-2 on both 22G and 26G RNAs could be indirect. Indeed, loss of 26G RNAs has also been observed in *mut-16* mutants (Zhang *et al*, 2011), suggesting that 26G RNA biogenesis may be coupled to 22G RNA biogenesis. Such interactions could also exist between RNAe initiation and maintenance mechanisms, and two of our findings hint into that direction. First, we noticed in our analysis of Tc1 activity that *pid-2;prg-1* double mutants display hardly any activity, while *pid-2* single mutants do. Second, we found that the 22G RNA levels of the 21U sensor are higher in *prg-1* mutants compared with wild type (Fig 2). Our interpretation of these two results is that HRDE-1-driven silencing becomes stronger in *prg-1* mutants, due to increased availability of 22G RNA resources. Obviously, such effects will have an impact on 22G RNA pools, clouding the primary effects of any factor that is studied, and it is not unlikely that the gain and loss of 22G RNAs seen in our mutants are related to such competitive effects between different pathways. Finally, on top of these mechanistic complications, developmental defects may further convolute phenotypes. We noticed feminization and masculinization phenotypes in our experiments, and a specific 21U RNA has been shown to act in sex determination (Tang *et al*, 2018). Such effects could affect, for instance, the 22G RNAs from ALG-3/-4 targets, as these targets are enriched for spermatogenesis-specific functions. Clearly, biochemical experiments will be required to define molecular functions of the newly identified, but also already known, small RNA pathway components.

### A function for the eTudor domains of PID-4 and PID-5

PID-4 and PID-5 were identified as robust PID-2 interactors. Both proteins contain an eTudor domain, and mutants lacking both PID-4 and PID-5 behave very similar to *pid-2* mutants, suggesting that PID-2 acts through PID-4 and PID-5. We do not know how the PID-4/-5 interactions with PID-2 are mediated. Given that PID-4 and PID-5 do not simultaneously interact with PID-2, and that the PID-4 and PID-5 eTudor domains are similar, these eTudor domains are good candidates bind to PID-2. Many eTudor domains recognize and bind symmetrically di-methylated arginine or lysine residues of partner proteins (Pek *et al*, 2012; Gan *et al*, 2019). However, sequence alignments suggest that the eTudor domains of PID-4 and PID-5 may not bind di-methylated arginines. Curiously, our IP-MS experiments on PID-2 and PID-4 revealed mild enrichment for PRMT-5, an enzyme known to symmetrically di-methylate arginines (Siomi *et al*, 2010; Pek *et al*, 2012). It is thus possible that PID-2, via PID-4, brings PRMT-5 activity into the small RNA systems of *C. elegans*, in order to modify arginines on other small RNA pathway components. Such potential substrates for PRMT-5 could be PRG-1, as well as HRDE-1 or one of the RdRPs, all containing several RG motifs.

### Potential roles for PID-2, PID-4 and PID-5 in granule dynamics

With the notion that our localization studies are based on fluorescent protein tags, which could in principle affect subcellular partitioning, we found that PID-2 on the one hand, and PID-4 and PID-5 on the other hand, are present in distinct perinuclear granules. PID-2 forms foci that are clearly distinct from P granules, but adjacent to them, and Wan *et al* (2020) could show that PID-2/ZSP-1 is present in Z granules. PID-4 and PID-5 are very close to, or within P granules, although we cannot exclude that PID-4 and PID-5 may define another, yet unknown kind of perinuclear compartment which would be closely associated with P granules. Higher-resolution microscopy, such as performed by Wan *et al* (2020), will be required to further resolve these localization issues.

It is interesting that PID-2 interacts with PID-4 and PID-5, and affects PID-4 localization, despite being in different granules. Possibly, these findings are related to the fact that PID-2/ZSP-1 is found at the surface of Z granules (Wan *et al*, 2020), where it may mediate interactions between Z granules and P granules. From this perspective, our data would also be consistent with the possibility that PID-4 and PID-5 may be found at the periphery of P granules, like PID-2 is in Z granules, and that PID-2, PID-4 and PID-5 provide a molecular interface between P and Z granules. This could explain the observed protein–protein interactions and may also provide a clue to why the combined loss of PID-4 and PID-5 affects Z granules differently than loss of PID-2. Wan *et al* (2020) show that loss of PID-2/ZSP-1

hardens Z granules, and this may well be related to increase in Z granule size that we and Wan *et al* (2020) describe for *pid-2* mutants. In *pid-4;pid-5* double mutants, PID-2 is still present and can maintain normal Z granule liquidity. This may prevent the growth of Z granules and contribute to the fact that Z granules disappear.

Clearly, these are all hypotheses that will need to be tested in future experiments. However, the simple fact that several proteins acting at different steps in the Mutator pathway are found enriched in different phase-separated structures (Batista *et al*, 2008; Wang & Reinke, 2008; Claycomb *et al*, 2009; Updike & Strome, 2010; Phillips *et al*, 2012; Ishidate *et al*, 2018; Wan *et al*, 2018; Wan *et al*, 2020; our own work) indicates that exchange of molecules between different granules is required to ensure silencing. Not much is known about such trafficking between adjacent phase-separated condensates, and our study identifies a set of three novel proteins that are excellent candidates to act in such processes.

### A role for X-prolyl aminopeptidase activity in 22G and 26G RNA pathways?

Next to an eTudor domain, PID-5 has an X-prolyl aminopeptidase domain. Based on the loss of key catalytic residues, it is likely catalytically inactive. What could be the function of such a protease-like domain? Potentially, PID-5 could use this domain to bind and lock proteins that carry a proline at position 2, without cleaving the most N-terminal amino acid. This would require a stable association of this catalytically dead X-prolyl aminopeptidase domain. We are not aware of studies assessing the stability of substrate–enzyme interactions of catalytically dead X-prolyl aminopeptidases. Another exciting hypothesis is that PID-5 could use its aminopeptidase domain to dimerize with the active X-prolyl aminopeptidase APP-1. Indeed, the catalytic domain of APP-1 is known to dimerize (Iyer *et al*, 2015), and we found APP-1 significantly enriched specifically in those IP-MS experiments in which PID-5 was enriched. We envisage that such heterodimerization could have two alternative functions. First, it could bring the enzymatic activity of APP-1 into PID-5-positive granules. Alternatively, PID-5 could inhibit the catalytic activity of APP-1, by preventing APP-1 homodimerization. In the first model, APP-1 would not be expected to be present in P or Z granules in *pid-5* mutants, whereas in the second model, the localization of APP-1 would be independent of PID-5. We are unfortunately not aware of studies describing whether APP-1 needs to dimerize to be active or not. Another issue that will need to be addressed is the identification of APP-1/PID-5 substrates. In this light, the identification of the Z granule-resident argonaute protein WAGO-4 in PID-5 IPs is intriguing, as this WAGO protein bears an APP-1-compatible N-terminus.

### Evolutionary considerations

The fact that PID-4 and PID-5 bind to PID-2 in a mutually exclusive manner could point at a regulatory interaction between these two proteins. For instance, an appealing hypothesis would be that PID-4 could act to modulate the amount of PID-5 that can bind to PID-2. Indeed, PID-5 is expressed in a much more restricted area in the germline than PID-4 (Fig 6B–E, G and H). However, both proteins could also have completely independent functions. In

this light, the following observation is of interest. The *pid-4* gene is positioned directly next to *app-1* in the *C. elegans* genome. A scenario in which *pid-5* was formed by a gene duplication event, in which *pid-4* and *app-1* became joined together, seems an interesting possibility. Indeed, *pid-5* orthologs are only present in some of the *Caenorhabditis* species (*C. remanei*, *C. brenneri*, *C. briggsae*) (Fig EV3D), but not, for instance, in *C. japonica*, which is evolutionary more distant from the above-mentioned species (Kanzaki *et al*, 2018). However, a *pid-4* ortholog is present in *C. japonica* and is also located next to *app-1*. This is consistent with the idea that a genomic rearrangement between *pid-4* and *app-1* may have happened in the last common ancestor of *C. elegans*, *C. remanei*, *C. brenneri* and *C. briggsae*, leading to the formation of *pid-5*. This implies that PID-4 has a PID-5 unrelated function in *C. japonica* and may still have this function in *C. elegans*. The fact that *pid-4* and *pid-5* single mutants do have phenotypes is consistent with this idea.

## Materials and Methods

### Strain maintenance

Worm strains have been grown according to standard laboratory conditions on NGM plates seeded with *Escherichia coli* OP50 and grown at 20°C, unless otherwise stated (Brenner, 1974). We used the N2 Bristol strain as wild-type strain. Strains used in this study are listed in the Appendix.

### Microscopy

20–25 worms have been picked to a drop of M9 (80 µl) on a slide, washed and then fixed with acetone (2 × 80 µl). After acetone has evaporated, worms have been washed 2 × 10 min with 80 µl of PBS-Triton X-100 0.1%. After removing the excess of PBS-Triton X-100 0.1%, the worms have been mounted on a coverslip with Fluoroshield™ with DAPI (5 µl) (Art. No. F6057, Sigma).

Alternatively, for live imaging, 20–25 worms have been picked to a drop of M9 (80 µl) on a slide, washed and then 2 µl of 1 M $NaN_3$ have been added to paralyse the worms. After removing the excess of M9, a slide prepared with 2% agarose (in water) has been placed on top of the coverslip and worms have been imaged directly.

Images have been acquired either at a Leica DM6000B microscope (objective HC PL FLUOTAR 20× 0.5 dry, Art. No. 11506503, Leica) or at a Leica TCS SP5 STED CW confocal microscope (objective HCX PL APO 'CS 63×/1.2 water UV', Art. No. 11506280, Leica). Images have then been processed with Leica LAS software and ImageJ. Images representing the expression of PID-2, PID-4, PID-5, and P and Z granules markers have been processed with the Huygens Remote Manager v3.6 and deconvoluted (Huygens Deconvolution, SVI).

For scoring the 21U sensor as active or silenced, we have used a Leica M165FC widefield microscope. The 21U sensor has been scored as active, if the fluorescence was easily visible with a lower magnification (Plan APO 1.0×, Art. No. 10450028; Leica); faint, if the fluorescence was only visible with a higher magnification (Plan APO 5.0×/0.50 LWD, Art. No. 10447243; Leica); and silenced, if no fluorescence was visible. The worms have been later used also for live imaging with a Leica DM6000B microscope as described above.

### Colocalization analysis

In order to perform colocalization analysis of the fluorescently tagged proteins PID-2, PID-4, PID-5 and of the P and Z granules markers, we used the DiAna plugin of (Fiji Is Just) ImageJ (Schindelin *et al*, 2012; Gilles *et al*, 2017). We used the deconvoluted images (Huygens Deconvolution, SVI), which consist of a single *z*-stack, and analysed the two fluorescence channels of interest at the time. First, we cropped an area containing 1–8 nuclei within the pachytene zone of the gonad arm, adding the area to the ROI manager, to ensure cropping of the same area in both channels being analysed. We then used DiAna_Segment to apply a mask to the images, considering all objects with size from 1 to 2,000 pixels and then adjusted the threshold to ensure a faithful segmentation of the images. After segmentation, we performed the analysis using DiAna_Analyse and measured the surface area ($\mu m^2$), the distance between the two signals ($\mu m$) and the number of objects present in the cropped area. We then represented the distance between centres as a measure of colocalization. For each couple of fluorescent proteins, we analysed 4–10 images of gonads from individual animals.

### Small RNA sequencing

### RNA extraction

Synchronized gravid adults have been collected with M9 and fast-frozen on dry ice in 250 µl of Worm Lysis Buffer (200 mM NaCl; 100 mM Tris–HCl pH = 8.5; 50 mM EDTA pH = 8; 0.5% SDS). 30 µl of Proteinase K (20 mg/ml; Art. No. 7528.1, Carl Roth) has been added to dissolve the worms for 90 min at 65°C with gentle shaking. Lysate has been centrifuged at maximum speed for 5 min at room temperature (RT), and the supernatant was transferred on a Phase Lock Gel tube (Art. No. 2302830, Quantabio). 750 µl TRIzol LS (Art. No. 10296028, Invitrogen™) has been added per 250 µl of sample, and after homogenization, the samples have been incubated for 5 min at RT to allow complete dissociation of the nucleoprotein complex. Then, 300 µl of chloroform (Art. No. 288306, Sigma-Aldrich) was added per 750 µl of TRIzol LS, and the samples were incubated for 15 min at room temperature after mixing. Samples have been centrifuged at 12,000 × *g* for 5 min at RT, and another round of chloroform extraction has been performed. The aqueous phase has been then transferred to an Eppendorf tube, and 500 µl of cold isopropanol was added to precipitate the RNA; samples have been mixed vigorously, incubated at RT for 10 min and spun down at maximum speed for at least 10 min at 4°C. The pellet was then washed twice with 1 ml of 75% ethanol and centrifuged for 5 min at 7,500 × *g* at 4°C. The pellet has been dried and diluted in 50 µl of nuclease-free water with gentle shaking for 10 min at 42 °C. In order to remove any contamination of genomic DNA, 5 µl of 10X TURBO™ DNase Buffer and 1 µl of TURBO™ DNase (Art. No. AM2238, Invitrogen™) were added to the RNA and incubated at 37 °C for 30 min with gentle shaking. The reaction has been stopped by adding 5 µl of 10× TURBO™ DNase Inactivation Reagent. Samples have been centrifuged at 10,000 × *g* for 90 s and RNA transferred to a fresh tube. RNA quality has been assessed at NanoDrop and on agarose gel, and then, samples have been further processed for enrichment of small RNA populations.

### Library preparation and sequencing

For each strain, three biological replicates have been used for RNA extraction and library preparation. RNA was treated with RppH (RNA 5′ pyrophosphohydrolase, Art. No. M0356S, New England Biolabs) to dephosphorylate small RNAs and specifically enrich for 22G RNAs, as previously described (Almeida *et al*, 2019b). For each sample, 1 µg of RNA was incubated for 1 h at 37°C with 5 units of RppH and 10× NEB Buffer 2. After dephosphorylation, 500 mM EDTA was added and samples were incubated for 5 min at 65°C to stop the RppH treatment and RNA was precipitated with sodium chloride/isopropanol. Small RNAs (15–30nt) were enriched by performing size selection on a 15% TBE-Urea polyacrylamide gel (Bio-Rad) prior to library preparation.

For strains, RFK184, RFK422, RFK764-RFK769 (full list of strains in Appendix) and small RNAs (15–30nt) were enriched by performing size selection of the RNA prior to library preparation. RNA samples were run on a 15% TBE-Urea polyacrylamide gel (Bio-Rad) and purified with sodium chloride/isopropanol precipitation. NGS Library Prep was performed with NEBNext Small RNA Library Prep Kit for Illumina following instructions of manual, with a modification of the adaptors, for which custom-made random barcodes for both 3′ SR adaptor and 5′ SR adaptor were used (HISS Diagnostics GmbH, 5′rApprnrnrnrnAGATCGGAAGAGCACACGTCT-NH2-3′, and 5′rGrUrUrCrArGrArArGrUrUrCrUrArCrArGrUrCCrGrArCrGrArUr Crnrnrnrn-3′, respectively). Libraries were profiled in a High Sensitivity DNA on a 2100 Bioanalyzer (Agilent Technologies), quantified using the Qubit dsDNA HS Assay Kit, in a Qubit 2.0 Fluorometer (Life Technologies), and sequenced on a HiSeq 2500 (Illumina).

For the other sequenced strains, NGS Library Prep was performed with NEXTFlex Small RNA-Seq Kit V3 following Step A to Step G of Bioo Scientifics' standard protocol (V16.06), using the NEXTFlex 3′ SR adaptor and 5′ SR adaptor (5′ rApp/NNNNTGGAATTCTCGGGTGCCAAGG/3ddC/ and 5′ GUUCAGA-GUUCUACAGUCCGACGAUCNNNN. Amplified libraries were purified by running a 8% TBE gel and size-selected for 15–40nt. Libraries were profiled in a High Sensitivity DNA Chip on a 2100 Bioanalyzer (Agilent Technologies), quantified using the Qubit dsDNA HS Assay Kit, in a Qubit 2.0 Fluorometer (Life Technologies), and sequenced on a NextSeq 500/550 (Illumina).

### Read procession and mapping

Before mapping to the reference sequences, reads were processed in the following manner: (i) trimming of sequencing adapters with cutadapt v1.9 (-a TGGAATTCTCGGGTGCCAAGG -a AGATCGGAA-GAGCACACGTCT -O 5 -m 26 -M 38) (Martin, 2011); (ii) removal of reads with low-quality calls with the FASTX-Toolkit v0.0.14 (fastq_quality_filter -q 20 -p 100 -Q 33); (iii) collapsing of PCR duplicates (custom bash script), making use of the unique molecule identifiers (UMIs) added during library preparation; (iv) trimming of UMIs with seqtk v1.2 (seqtk trimfq -b 4 -e 4); and (v) removal of very short sequences with seqtk v1.2 (seqtk seq -L 15). Read quality was assessed before and after these processing steps with FastQC v0.11.5 (http://www.bioinformatics.babraha m.ac.uk/projects/fastqc).

Reads that passed the above filtering steps were mapped to a custom *C. elegans* genome (WBcel235) to which the 21U sensor sequence (Bagijn *et al*, 2012) was added as an extra contig. The mapping was done with bowtie v0.12.8 (-q –sam –phred33-quals –tryhard –best –strata –chunkmbs 256 -v 0 -M 1) (Langmead *et al*, 2009). Reads mapping to structural genes were filtered out (rRNA/

tRNA/snoRNA/snRNA) using Bedtools 2.25.0 (bedtools intersect -v -s -f 0.9), and further analysis was performed using non-structural RNAs. To generate genome browser tracks, we used a combination of Bedtools v2.25.0 (genomeCoverageBed -bg -split -scale -ibam -g) (Quinlan & Hall, 2010), to summarize genome coverage normalized to mapped non-structural reads (rRNA/tRNA/snoRNA/snRNA) * 1 million (RPM, reads per million), and bedGraphToBigWig to finally create the bigwig tracks. More detailed information is available in the Appendix.

## Transposon excision analysis

For each analysed genotype, mutant worms carrying the *unc-22::Tc1 (st136)* insertion were singled into a 10 cm NGM plate seeded with 100 µl of OP50 and grown at 20°C. Plates were regularly checked for reversion events until they were starved. When starved, the population was estimated at 10,000 animals. Average animal populations on non-starved plates were estimated by counting sectors of a number of plates. Transposition frequencies at each time point were calculated using the following formula: $f = -\ln[(T - R)/T]/N$, where $T$ = total number of plates scored, $R$ = number of plates with revertants and $N$ = number of worms on the plate. A table with the numbers used to make the graph in Fig 1C is given in the Appendix Materials and Methods.

## Mortal germline assay

Before starting the experiment, mutants have been outcrossed four times. For the assay, N2 has been used as wild-type strain and the desired mutant strains have been tested. For each strain, 6 L3 worms have been picked onto 15 NGM plates (10 cm$^2$) seeded with 300 µl OP50, grown at 25°C and followed over time. Worms have been picked every 4–6 days, before starvation, and we assumed 2–3 generations, respectively, have passed. Worms have been passed to fresh plates every 4–6 days until all the mutants died.

## Production of PID-2 protein for antibody generation

Information available in the Appendix.

## Transgenic line generation

Information available in the Appendix.

## Generation of mutant and endogenously tagged lines using CRISPR/Cas9 technology

### Cloning

All the sgRNAs have been cloned in the vector p46169 [a gift from John Calarco (Friedland *et al*, 2013)], except for Y45G5AM.2_sgRNA7, which has been cloned in the vector pRK2412 [pDD162 backbone, Cas9 deleted with improved sgRNA(F + E) sequence, as described in Chen *et al* (2013)].

### Generation of mutant lines

Wild-type worms have been injected with an injection mix containing 50 ng/µl pJW1259 (encoding for *Peft-3::cas9::tbb-2 3'UTR*, a gift from Jordan Ward (Ward, 2014)), co-injection markers (10 ng/µl

pGH8; 5 ng/µl pCFJ104; 2.5 ng/µl pCFJ90) and 30 ng/µl of each of the plasmids encoding for the sgRNAs. We isolated two deletion alleles of *Y45G5AM.2/pid-5* (*xf181* and *xf182*) and two deletion alleles of *W03G9.2/pid-4* (*xf184* and *xf185*). Each allele has been sequenced to pinpoint the exact deletion at nucleotide resolution. The mutant strains have been outcrossed two times against wild-type N2 strain to remove any potential off-target effect of Cas9 and used for further experiments.

### Generation of endogenously tagged lines

In order to introduce an epitope tag at endogenous loci, we used the co-conversion approach as previously described (Arribere *et al*, 2014). After injections, worms have been kept at 20°C and F1 offspring with a roller phenotype (*rol-6*) were singled out. The tagged strains have been outcrossed two times against wild-type N2 strain to remove any potential off-target effect of Cas9 and used for further experiments.

To introduce a fluorescent protein at the endogenous locus, we have first used a *unc-58* co-conversion approach (Arribere *et al*, 2014) to introduce a sequence of 20 nucleotides of *dpy-10* gene that serves as efficient protospacer sequence for subsequent edits, as previously described (El Mouridi *et al*, 2017). The tagged strains were sequenced and were outcrossed two times against wild-type N2 strain to remove any potential off-target effect of Cas9 and used for further experiments.

Sequences and more detailed information on procedures are available in the Appendix

## Protein extraction and immunoprecipitation

Information available in the Appendix.

## Mass spectrometry

Label-free quantitative mass spectrometry has been performed as described in Ref. (Almeida *et al*, 2018). After boiling (see above), the samples were separated on a 4–12% gradient Bis-Tris gel (NuPAGE Bis-Tris gels, 1.0 mm, 10 well; Art. No. NP0321; Life Technologies) in 1× MOPS (NuPAGE 20× MOPS SDS running buffer; Art. No. NP0001; Life Technologies) at 180 V for 10 min, afterwards processed by in-gel digest (Shevchenko *et al*, 2007; Bluhm *et al*, 2019) and desalted using a C18 StageTip (Rappsilber *et al*, 2007). The digested peptides were separated on a 25-cm reverse-phase capillary (75 µM inner diameter) packed with Reprosil C18 material (Dr. Maisch) with a 2-h gradient from 2 to 40% Buffer B (see StageTip purification) with the EASY-nLC 1,000 system (Thermo Scientific). Measurement was done on a Q Exactive Plus Mass Spectrometer (Thermo Scientific) operated with a Top 10 data-dependent MS/MS acquisition method per full scan (Bluhm *et al*, 2016). The measurements were processed with the MaxQuant software, version 1.5.2.8 (Cox & Mann, 2008) against the Wormbase *C. elegans* database (version of WS265) for quantitation and the Ensemble *E. coli* REL606 database (Version Oct 2018) to filter potential contaminations.

## Protein extraction and Western blot

Information available in the Appendix.

## Data availability

The datasets produced in this study are available in the following databases:

- RNA-Seq data: Sequence Read Archive BioProject ID PRJNA612883 (https://www.ncbi.nlm.nih.gov/bioproject/PRJNA612883)
- Mass spectrometry data: PRIDE PXD018402 (http://www.ebi.ac.uk/pride/archive/projects/PXD018402)

**Expanded View** for this article is available online.

## Acknowledgements

We thank members of the Ketting Lab for stimulating discussions. We thank Walter Bronkhorst, Joana Costa Pereirinha and Roberto Orrù for valuable technical assistance. We also thank the IMB Core facilities Media Lab, Microscopy, Bioinformatics, Protein production and Genomics for excellent experimental and technical support. MP was supported by a Boehringer Ingelheim Fonds PhD Fellowship. This work was further supported by grants, KE1888/1-1 and KE1888/1-2, from the Deutsche Forschungsgemeinschaft (RFK), and by a grant from Fundação para a Ciência e Tecnologia ([FCT]SFRH/BD/51001/2010; BFMA). Open Access funding enabled and organized by Projekt DEAL.

## Author contribution

MP, AMJD and RFK planned experiments. MP generated strains, and performed genetics, microscopy and molecular biology. RFK and MP performed Tc1 reversion assays. AMJD analysed sequencing data. JS generated strains and performed microscopy. SH generated strains. BFMA identified *pid-2* and performed initial analyses. SD and FB performed quantitative mass spectrometry. All authors were involved in discussion of the data. MP and RFK wrote the manuscript with input from all authors.

## Conflict of interest

The authors declare that they have no conflict of interest.

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
