## [Review Process File · The EMBO Journal]

Intrinsically disordered protein PID-2 modulates Z granules and is required for heritable piRNA-induced silencing in the *C. elegans* embryo

Maria Placentino, Antonio de Jesus Domingues, Jan Schreier, Sabrina Dietz, Svenja Hellmann, Bruno de Albuquerque, Falk Butter, and Rene Ketting
DOI: [10.15252/embj.2020105280](https://doi.org/10.15252/embj.2020105280)

Corresponding author(s): Rene Ketting (r.ketting@imb-mainz.de)

Review Timeline:

Submission Date:	14th Apr 20
Editorial Decision:	15th May 20
Revision Received:	25th Aug 20
Editorial Decision:	18th Sep 20
Revision Received:	25th Sep 20
Accepted:	2nd Oct 20

Editor: Stefanie Boehm

Transaction Report:

Thank you for submitting your manuscript characterizing three additional factors involved in *C. elegans* small RNA silencing for consideration by The EMBO Journal. We have now received three referee reports on your study, which are included below for your information.

As you will see, the reviewers are overall positive and acknowledge the thorough characterization of PID-2, PID-4 and PID-5. Nonetheless they also raise some concerns that would need to be addressed in a revised manuscript. In particular, both referee #1 and referee #3 indicate recent studies that should be taken into account. In addition to discussing the respective work, they suggest potentially assessing the interaction of PID-2 with SIMR-1, as well as determining if small RNAs against histones are present in the mutants. Both points can likely be addressed by re-analyzing available data and should at least be attempted and discussed. In addition, referee #2 and referee #3 suggest that the section of the manuscript describing the genetic experiments of figure 1 and 2 should be revised to make the results and conclusions more clear to the reader, as well as providing quantification of the reporter in figure 1 (referee #3). Moreover, referee #1 raises several questions regarding the proteomic experiments and their analysis, and is also not convinced by the conclusion regarding the production of 22g RNA from 5' ends (Fig. 3C). These concerns should be addressed. Please also carefully respond to all other issues raised by the referees and revise the manuscript and figures as applicable.

Please note that it is our policy to allow only a single round of major revision. We realize that lab work worldwide is currently affected by the COVID-19/SARS-CoV-2 pandemic and that an experimental revision may be impaired or delayed. We can extend the revision time when needed, and we have extended our 'scooping protection policy' to cover the period required for a full revision. However, it is nonetheless important to clarify any questions and concerns at this stage, and we encourage you to discuss a revision plan and any potential issues you may foresee with us as soon as possible.

Please also feel free to contact me should you have any other further questions. Thank you for the opportunity to consider your work for publication. I look forward to receiving your revised manuscript.

Referee #1:

In this manuscript, Placentino et al. have identified three novel factors called PID-2, PID-4 and PID-5 that participate in the initiation of piRNA-mediated silencing. PID-2 has been found in a screening for factors that desilenced a piRNA reporter. PID-4 and PID-5 have been found through proteomic approaches in PID-2 IPs. The authors also show that mutation in these factors affect the accumulation of 26G RNAs and 22G-RNAs belonging to the mutator targets. Moreover, the three proteins localize in perinuclear foci near Z- and P-granules. Mutation of pid-2 or pid-4;pid-5 double mutant shows reduced Z granules compared to P-granules. Overall, the experiments are well executed and globally support their conclusions. However, in my opinion the study remains a bit descriptive, because it lacks a mechanistic understanding of how these three new factors are specifically involved in the initiation of piRNA silencing and how they affect subpopulation of 22G-RNAs. Moreover, mutations in these factors do not completely abolish the capacity of piRNA silencing and the effect observed on Z granule is quite mild. Also, some emphasis is given to the possible role of N-terminal proteolysis in RNAe, based on the interaction between PID-5 and APP-1. However, no functional experiments are shown to implicate N-terminal proteolysis in RNAe.

Specific comments

- In their proteomic experiments the authors only listed few proteins of interest. However, there are much more significant enriched proteins ($p > 0.05$). Therefore, the authors should list in a table all the significant interactors and maybe identify some categories to help the reader find some other relevant interactors. For instance, did they find any other RNAi of germ granule factors in their IPs? Also, they should list all the proteins that have been found in common to all the different IPs of PID-2 as well as common interactors with PID-4 and PID-5. In addition, they need to decide if they want to focus only on significant interactors. In some plots non-significant interactors (such as APP-1 and PRMT-5) are shown. PRMT-5 is also non-significantly enriched in PID-4 IPs. Therefore, I am not sure they can claim that this interaction is real if not confirmed by reciprocal IP and mass spec or Co-IP experiments.
- The authors claim that "PID-2 is required to specifically maintain 22G RNA production from the 5' parts of the target transcripts, suggesting a role in RdRP processivity." In my opinion this is not substantiated based on Figure 3C. The small decreased at the 5' might not be significant. Maybe to substantiate this claim they need to generate Metaprofile of genes with upregulated 22G-RNAs and downregulated 22G-RNAs and see whether they see a much dramatic effect on the 5' end of transcripts.
- In proteomics data a special emphasis has been placed on interaction of APP-1 in addition to pid-4 and pid-5, however, it is not clear what is the functional significance. In the abstract they wrote "PID-5 has a domain related to the X-prolyl aminopeptidase protein APP-1, and binds APP-1, implicating N-terminal proteolysis in RNAe." There is no support for a role of APP-1 in RNAe at the moment. They can at least test the role of APP-1 in RNAe using the same type of assay they have used for pid-2.
- A recent paper by Manage et al, 2020 have shown another tudor domain protein, called SIMR-1 in piRNA silencing. They have shown that it localizes in a distinct perinuclear region between mutator and P-granule. Maybe the author should look whether their identified factors colocalize with SIRM-1 (is there any SIMR-1 enrichment in the proteomic data?).
- In the discussion the authors state "However, the fact that all three identified PID proteins affect

Z granule formation, without affecting ZNFX-1 protein levels, strongly supports a role in Z granule homeostasis." Given the reduction in the Z-granule, shouldn't the Authors expect an impact on heritable RNAi? Similarly, why pid-2 mutant doesn't show the same redistribution of small RNAs from 3' to 5' that occurs in znfx-1 mutant?

Referee #2:

Ketting and colleagues characterize a gene encoding an intrinsically disordered protein that they previously uncovered in a genetic screen for factors that desilence a piRNA sensor. In doing so, they identify a pair of genes encoding functionally overlapping Tudor domain proteins. Using some impressive genetics along with small RNA sequencing, they reveal a role for the 3 factors in establishing siRNA formation during embryogenesis, possibly by promoting RNA-dependent RNA polymerase activity at target transcripts or perhaps in orchestrating other proteins in the pathway. Interestingly the PID proteins mingle in germ granules but occupy somewhat distinct domains. These factors may represent a missing link between primary and secondary small RNA formation and as such provide an exciting advance that will likely have broad implications. A few minor suggestions, mostly related to clarity, are noted below.

Minor Points

I'm not sure if there's anyway to resolve this without removing some of non-essential data, but the sections describing Figures 1-2 and S1-S2 are very dense with all the complex genetics. Once you get past that these sections it becomes much more enjoyable to read.

Figure S1E. A key describing the symbols next to the genotypes in S1E would be helpful.

Line 275. It seems that ergo-1-dependent 22G-RNA are reduced in only one of the two pid-2 alleles.

Figure 4B. To me it would make more sense to have N2 on the right so that you're looking at positive enrichment as opposed to negative enrichment but perhaps the way its presented is the standard approach.

Line 350. While I don't think its necessary, and I'm not suggesting that you do it as it's probably not justified by the time and expense, I am curious why you didn't sequence small RNAs from the pid-4 pid-5 double mutant as given their redundancy that would seem to make more sense than examining each individually. I follow your logic but it still strikes me as a bit odd. But again I don't think it would add enough value to the manuscript to justify additional experiments.

Figures 5G-5H. The generational fertility assays reported were done at 25C but I'm curious if you also did the assays at 20C, as the temperature sensitivity of the Mrt phenotype seems to distinguish certain factors, such as henn-1 (25C only) and prg-1 (20C and 25C). These assays take a very long time and I'm certainly not suggesting that the manuscript should be delayed by additional experiments but if you happened to have already done it, I believe it's worth reporting even if you view it as a negative result.

Figures 7A-7C. Size-matching the magnification in the insets in these images would make the comparison more straightforward.

It's interesting that PID-2 and PID-4/PID-5 don't colocalize despite being in a complex with one

another. Perhaps you could speculate on this in the discussion where you talk about germ granules. Do you think the interaction between the proteins is transient?

There seems to be some gaps in the methods describing small RNA library preparation.

Referee #3:

In this manuscript Placentino et al. identify and characterize three novel components of the piRNA pathway in *C. elegans*, PID-2, -4 and -5. The piRNA pathway and other small RNA silencing pathways are essential for many aspects of biology, with clearly important roles in the germline of diverse animals. The authors provide a thorough analysis of these new components, in the context of the multiple small RNA pathways that operate in the germline of *C. elegans*.

The Ketting lab had previously isolated an allele of *pid-2* in a forward mutagenesis screen for genes involved in silencing of a reporter for piRNA-induced silencing. The PID-2 protein does not have identifiable domains but the authors use diverse tools to understand its function:

- In the first part of the manuscript, Placentino et al. describe the effects of loss-of function of *pid-2* (alone or in combination with other mutants in the pathway) on a variety of assays for different aspects of piRNA-mediated silencing: they show that PID-2 is necessary for robust silencing of the piRNA-activity reporter and of endogenous transposons. Interestingly, PID-2 is also necessary for establishment of de novo silencing of the reporter (based on maternally-deposited piRNAs). The authors also use small RNA sequencing to uncover the molecular consequences of loss of PID-2: *pid-2* mutant animals have changes in different small RNA populations and as expected, specifically fail to accumulate small RNAs against the sensor.

- In the second part of the manuscript, the authors identify PID-4 and PID-5 as interactors of PID-2, using mass spectrometry. They further identify interactors of PID-4 and PID-5 retrieving not only PID-2 but also other proteins that are potentially relevant for their function. Using a similar genetic analysis and small RNA analysis as for *pid-2*, the authors go on to show that PID-4 and PID-5 are also involved in piRNA-mediated silencing, with partially overlapping functions. All three proteins contribute to the immortality of the germline.

- Finally, the authors analyze the localization of all three proteins in the *C. elegans* germline, in the context of different structures that are known to be involved in small RNA biology: RNA and protein condensates called P- or Z-granules. PID-2 is in granules adjacent to P-granules (likely Z-granules) while PID-4 and -5 are found in granules that are very close to (and in some instances seem to co-localize with some) P-granules and Z-granules. All three proteins are required for Z-granule properties, PID-2 for size and all three for Z-granule number.

Overall, this is a thorough, high-quality analysis of these new components of the pathway. This work will be of high interest to the relatively broad community studying small RNA pathways in diverse systems. I have relatively minor suggestions for improvement of the manuscript, in order of importance:

1. The main need for improvement in my opinion is the initial presentation of the effect of *pid-1* mutation on the sensor. This is now currently presented over Figures 1B, S1C and S1E. Together these figures somewhat provide an idea of how penetrant the effect is (number of animals in which the sensor is de-silenced) and the magnitude of the de-silencing (currently measured by qPCR and the quantification of one image per genotype). This is important information that should be presented clearly in the text and in a main figure; figure 1B now just shows one image (which is unclear how representative it is given the quantification in the supplemental figure). Ideally, the

authors could provide a panel in the main figure with a quantitative analysis of images such as the one shown in 1B over many animals. Plotting these data would show in a single plot the penetrance and magnitude of the effect. Moreover, the effect of *pid-1* on the epiallele of the sensor that is epigenetically silenced is also presented in a rather convoluted manner. First, based on the one image shown in 1B the authors say there's no effect of epigenetic silencing, but then based on the qPCR analysis in S1E there is an effect. It is unclear if the authors actually quantified the sensor based on micrographs such as the one in figure 1B. If the authors properly quantified the sensor under these conditions and the effects do not match, they should provide an explanation for this. Again, ideally the authors could provide a quantification of the reporter over many individual animals and this could resolve the seeming discrepancy.

2. The authors describe the mortal germline phenotype of *pid-2*, *-4*, *-5* mutant animals and suggest it's related to the mortal germline phenotype in *Piwi* (*prg-1*) mutants, but don't cite a recent publication by Barucci et al, Nat Cell Biol 2020, that shows that *prg-1* mutant animals become progressively sterile due to the ectopic targeting of histone mRNAs. I think this is relevant work that should be cited, but in addition, it would make a lot of sense for the authors to check in their small RNA sequencing data whether they also observe small RNAs against histones. This should be a relative straightforward analysis that would further help place their observations in the context of what is known about the pathway.

3. In Figure 1C, the level of Tc1 reversion in *hrde-1* single-mutants is missing for correct interpretation of the contribution of PID-2.

4. *MosSCI* and *MiniMos* alleles should be designated *xfSi####* rather than *xfIs####* to denote single copy integration.

5. In figure 7, the order in which the panels are arranged in the figure doesn't coincide with the text, it would be easier for the reader if the authors swapped 7 D,E and G,H.

Dear Stefanie, dear reviewers,

we would like to thank all three reviewers for their thoughtful comments, and good suggestions for further improvement of our manuscript. We also appreciate the opportunity to revise our manuscript for The EMBO Journal. Below, we will address the raised issues one-by-one, and indicate how we have changed our manuscript in response to the raised issues. The original reviewer comments are shaded in grey, the answer follows in regular text. First, however, we want to mention that we have changed most of the plots related to 22G RNA analysis, since we found a small bug in our analysis, leading to RPM values of more than 1M. This was related to a split calling of 22G RNA reads and the pool of non-structural reads. Now, we first define the non-structural reads, and from these we call 22G RNAs. In addition, we now are more strict in 22G RNA definition. In the original submission we took all reads between 20 and 23 nucleotides, now we ask for a 5'G to be present as well. This leads to a cleaner pool of 22G RNAs. None of the conclusions are affected by these changes, but the absolute numbers in the graphs did change. We have adapted the Methods section to reflect these changes in analysis.

Response to reviewer comments.

Referee #1:

In this manuscript, Placentino et al. have identified three novel factors called PID-2, PID-4 and PID-5 that participate in the initiation of piRNA-mediated silencing. PID-2 has been found in a screening for factors that desilenced a piRNA reporter. PID-4 and PID-5 have been found through proteomic approaches in PID-2 IPs. The authors also show that mutation in these factors affect the accumulation of 26G RNAs and 22G-RNAs belonging to the mutator targets. Moreover, the three proteins localize in perinuclear foci near Z- and P-granules. Mutation of pid-2 or pid-4;pid-5 double mutant shows reduced Z granules compared to P-granules. Overall, the experiments are well executed and globally support their conclusions. However, in my opinion the study remains a bit descriptive, because it lacks a mechanistic understanding of how these three new factors are specifically involved in the initiation of piRNA silencing and how they affect subpopulation of 22G-RNAs. Moreover, mutations in these factors do not completely abolish the capacity of piRNA silencing and the effect observed on Z granule is quite mild. Also, some emphasis is given to the possible role of N-terminal proteolysis in RNAe, based on the interaction between PID-5 and APP-1. However, no functional experiments are shown to implicate N-terminal proteolysis in RNAe.

Reply:

We agree that our current data do not provide a clear molecular mechanism for how the three new PID-proteins act. However, we do not believe that this is something that can and has to be resolved at this stage. Before mechanisms can be resolved, components need to be identified, and roughly placed into the RNAe scheme. We note that for instance for ZNFX-1, and for many other RNAi/RNAe components clear molecular mechanisms are still lacking. That does not mean that the studies that identified them have less of an impact. It shows this field is overall not yet at a point where such mechanisms can really be resolved. Often, mechanism is inferred from interactions, such as the interaction between EGO-1 and ZNFX-1. Similarly, we believe our identification of APP-1 in our IPs is of relevance, even if we do not yet study that factor in detail. Obviously, studies on APP-1 are scheduled, and we are convinced these will provide provocative data for a next publication. Finally, we note that the three proteins we identify are the first that effect Z granule homeostasis, and as such open up studies into the function of this germ granule.

Overall, despite these limitations, we believe our identification of three novel proteins, and their effects on granules has significant impact, and fits The EMBO Journal very well.

Specific comments

- In their proteomic experiments the authors only listed few proteins of interest. However, there are much more significant enriched proteins ($p > 0.05$). Therefore, the authors should list in a table all the significant interactors and maybe identify some categories to help the reader find some other relevant interactors. For instance, did they find any other RNAi of germ granule factors in their IPs? Also, they should list all the proteins that have been found in common to all the different IPs of PID-2 as well as common interactors with PID-4 and PID-5. In addition, they need to decide if they want to focus only on significant interactors. In some plots non-significant interactors (such as APP-1 and PRMT-5) are shown. PRMT-5 is also non-significantly enriched in PID-4 IPs. Therefore, I am not sure they can claim that this interaction is real if not confirmed by reciprocal IP and mass spec or Co-IP experiments.

We now provide the requested information in a supplemental excel file (Table EV2). We note also that the significance curves in the volcano plots were somewhat misleading. PRMT-5, and also some other factors were in fact significantly enriched according to our cut-offs, but fell below the significance curve due to somewhat arbitrary setting that define the curve of the significance line. We have adapted these curves. The significance of these interactors is clear, especially considering that we repeatedly detect them with different antibodies versus different proteins.

- The authors claim that "PID-2 is required to specifically maintain 22G RNA production from the 5' parts of the target transcripts, suggesting a role in RdRP processivity." In my opinion this is not substantiated based on Figure 3C. The small decreased at the 5' might not be significant. Maybe to substantiate this claim they need to generate

Metaprofile of genes with upregulated 22G-RNAs and downregulated 22G-RNAs and see whether they see a much dramatic effect on the 5' end of transcripts.

We now performed the analysis with target genes separated into those that gain or lose 22G RNAs. Genes that lose 22Gs lose them all over the gene body. Those that gain still lose them from the 5' part. Significance was tested by dividing the gene bodies into 10% bins. The 5' effect is highly significant, and is also found in *pid-4;pid-5* double mutant 22G RNA sequencing data. These new analyses are shown in Figures S2 and EV4. We now focused on Mutator and CSR-1 target genes only (the classes where we saw an effect), in order to not overload the figures with plots.

*- In proteomics data a special emphasis has been placed on interaction of APP-1 in addition to *pid-4* and *pid-5*, however, it is not clear what is the functional significance. In the abstract they wrote "PID-5 has a domain related to the X-prolyl aminopeptidase protein APP-1, and binds APP-1, implicating N-terminal proteolysis in RNAe." There is no support for a role of APP-1 in RNAe at the moment. They can at least test the role of APP-1 in RNAe using the same type of assay they have used for *pid-2*.*

As discussed above, we will not provide additional data on APP-1, as this in itself will provide sufficient information to constitute an entire paper. We have softened the wording in the abstract.

- A recent paper by Manage et al, 2020 have shown another tudor domain protein, called SIMR-1 in piRNA silencing. They have shown that it localizes in a distinct perinuclear region between mutator and P-granule. Maybe the author should look whether their identified factors colocalize with SIMR-1 (is there any SIMR-1 enrichment in the proteomic data?).

SIMR-1 was not enriched in our mass spec experiments. We note this now in the results section. We also checked whether SIMR-1 colocalized with PID-4 and PID-5, but did not find clear colocalization (new Figure EV5).

*- In the discussion the authors state "However, the fact that all three identified PID proteins affect Z granule formation, without affecting ZNFX-1 protein levels, strongly supports a role in Z granule homeostasis." Given the reduction in the Z-granule, shouldn't the Authors expect an impact on heritable RNAi? Similarly, why *pid-2* mutant doesn't show the same redistribution of small RNAs from 3' to 5' that occurs in *znfx-1* mutant?*

Yes, we in fact described an effect on stability of RNAe in *pid-2* and *pid-4/-5* mutants, implying an effect on inheritance. Apparently, this was not very clear. Also, the effect that 22G levels derived from the sensor drop, also when it is in an RNAe state indicates that inheritance is affected. We have re-written the first paragraphs of the results section to improve this, and we have added information to Figure 1. We also have changed the first paragraph of the discussion to make this more clear.

As to why *pid-2* mutants do not show the same effect as *znfx-1* mutants, we simply do not know. The fact that a gene has an effect on Z-granule homeostasis, however, does not imply that its phenotype has to be the same as that following loss of one of the Z-granule components. In fact, ZNFX-1 was still present at normal levels, as we show in Figure S6, clearly making the situation in these *pid* mutants different from that in *znfx-1* mutants. Finally, as we have no clear picture of what ZNFX-1 is doing, like we do not know what PID-2 is doing, there is currently no way to predict what we can expect from loss of any of these factors. For instance, PID-2 could in some way inhibit ZNFX-1 function, leading to overactivity in *pid-2* mutants. Given that many different models could be envisaged, we prefer to not discuss that in more detail.

Referee #2:

Ketting and colleagues characterize a gene encoding an intrinsically disordered protein that they previously uncovered in a genetic screen for factors that desilence a piRNA sensor. In doing so, they identify a pair of genes encoding functionally overlapping Tudor domain proteins. Using some impressive genetics along with small RNA sequencing, they reveal a role for the 3 factors in establishing siRNA formation during embryogenesis, possibly by promoting RNA-dependent RNA polymerase activity at target transcripts or perhaps in orchestrating other proteins in the pathway. Interestingly the PID proteins mingle in germ granules but occupy somewhat distinct domains. These factors may represent a missing link between primary and secondary small RNA formation and as such provide an exciting advance that will likely have broad implications. A few minor suggestions, mostly related to clarity, are noted below.

Thank you very much for the kind comments.

Minor Points

I'm not sure if there's anyway to resolve this without removing some of non-essential data, but the sections describing Figures 1-2 and S1-S2 are very dense with all the complex genetics. Once you get past that these sections it becomes much more enjoyable to read.

We apologize for the fact that this part of the manuscript was not written very clearly. We have re-organized this section to improve clarity. We hope this section is now more transparent and easier to read.

Figure S1E. A key describing the symbols next to the genotypes in S1E would be helpful.

This has been added (now this is Figure EV1A).

Line 275. It seems that ergo-1-dependent 22G-RNA are reduced in only one of the two pid-2 alleles.

Thank you for the sharp observation. This has been corrected in the text.

Figure 4B. To me it would make more sense to have N2 on the right so that you're looking at positive enrichment as opposed to negative enrichment but perhaps the way its presented is the standard approach.

We understand the point that is raised. However, we kept wild-type consistently on the left side, and mutant/tags on the right and we prefer to keep it this way. Indeed that means the enrichment versus mutants goes to the left, while in tagged strains it goes to the right.

Line 350. While I don't think its necessary, and I'm not suggesting that you do it as it's probably not justified by the time and expense, I am curious why you didn't sequence small RNAs from the pid-4 pid-5 double mutant as given their redundancy that would seem to make more sense than examining each individually. I follow your logic but it still strikes me as a bit odd. But again I don't think it would add enough value to the manuscript to justify additional experiments.

We decided to add double mutant data, despite the kind suggestion this would not be necessary. The results nicely confirm the idea that PID-4 and PID-5 are partially redundant, as the small RNA defects in *pid-4;pid-5* mutants is very similar to that observed in *pid-2* mutants, including which genes gain or lose 22G RNAs and the mild 5' specific effect on 22G RNAs. This double mutant data analysis is shown in Figure 5, EV4, S3 and S5.

Figures 5G-5H. The generational fertility assays reported were done at 25C but I'm curious if you also did the assays at 20C, as the temperature sensitivity of the Mrt phenotype seems to distinguish certain factors, such as henn-1 (25C only) and prg-1 (20C and 25C). These assays take a very long time and I'm certainly not suggesting that the manuscript should be delayed by additional experiments but if you happened to have already done it, I believe it's worth reporting even if you view it as a negative result.

We do not have 20C data unfortunately. Given that the temperature effect is hard to interpret anyway, we feel we should not invest time in this aspect at this moment.

Figures 7A-7C. Size-matching the magnification in the insets in these images would make the comparison more straightforward.

We have corrected this.

It's interesting that PID-2 and PID-4/PID-5 don't colocalize despite being in a complex with one another. Perhaps you could speculate on this in the discussion where you talk about germ granules. Do you think the interaction between the proteins is transient?

We now address this finding in the discussion. This may very well relate to the finding by the Kennedy lab that PID-2/ZSP-1 appears to be on the surface of Z granules, where it may interact with components of other granules. We do not wish to comment on the nature of interaction, as this really requires biochemistry. Fact is that we find them in IP-MS, indicating that the interactions are rather robust.

There seems to be some gaps in the methods describing small RNA library preparation.

Thanks you for noticing this. This has been corrected.

Referee #3:

In this manuscript Placentino et al. identify and characterize three novel components of the piRNA pathway in C. elegans, PID-2, -4 and -5. The piRNA pathway and other small RNA silencing pathways are essential for many aspects of biology, with clearly important roles in the germline of diverse animals. The authors provide a thorough analysis of these new components, in the context of the multiple small RNA pathways that operate in the germline of C. elegans.

The Ketting lab had previously isolated an allele of pid-2 in a forward mutagenesis screen for genes involved in silencing of a reporter for piRNA-induced silencing. The PID-2 protein does not have identifiable domains but the authors use diverse tools to understand its function:

- In the first part of the manuscript, Placentino et al. describe the effects of loss-of function of pid-2 (alone or in combination with other mutants in the pathway) on a variety of assays for different aspects of piRNA-mediated silencing: they show that PID-2 is necessary for robust silencing of the piRNA-activity reporter and of endogenous transposons. Interestingly, PID-2 is also necessary for establishment of de novo silencing of the reporter (based on maternally-deposited piRNAs). The authors also use small RNA sequencing to uncover the molecular

consequences of loss of PID-2: *pid-2* mutant animals have changes in different small RNA populations and as expected, specifically fail to accumulate small RNAs against the sensor.

- In the second part of the manuscript, the authors identify PID-4 and PID-5 as interactors of PID-2, using mass spectrometry. They further identify interactors of PID-4 and PID-5 retrieving not only PID-2 but also other proteins that are potentially relevant for their function. Using a similar genetic analysis and small RNA analysis as for *pid-2*, the authors go on to show that PID-4 and PID-5 are also involved in piRNA-mediated silencing, with partially overlapping functions. All three proteins contribute to the immortality of the germline.

- Finally, the authors analyze the localization of all three proteins in the *C. elegans* germline, in the context of different structures that are known to be involved in small RNA biology: RNA and protein condensates called P- or Z-granules. PID-2 is in granules adjacent to P-granules (likely Z-granules) while PID-4 and -5 are found in granules that are very close to (and in some instances seem to co-localize with some) P-granules and Z-granules. All three proteins are required for Z-granule properties, PID-2 for size and all three for Z-granule number.

Overall, this is a thorough, high-quality analysis of these new components of the pathway. This work will be of high interest to the relatively broad community studying small RNA pathways in diverse systems. I have relatively minor suggestions for improvement of the manuscript, in order of importance:

We thank the reviewer for the kind words.

1. The main need for improvement in my opinion is the initial presentation of the effect of *pid-1* mutation on the sensor. This is now currently presented over Figures 1B, S1C and S1E. Together these figures somewhat provide an idea of how penetrant the effect is (number of animals in which the sensor is de-silenced) and the magnitude of the de-silencing (currently measured by qPCR and the quantification of one image per genotype). This is important information that should be presented clearly in the text and in a main figure; figure 1B now just shows one image (which is unclear how representative it is given the quantification in the supplemental figure). Ideally, the authors could provide a panel in the main figure with a quantitative analysis of images such as the one shown in 1B over many animals. Plotting these data would show in a single plot the penetrance and magnitude of the effect. Moreover, the effect of *pid-1* on the epiallele of the sensor that is epigenetically silenced is also presented in a rather convoluted manner. First, based on the one image shown in 1B the authors say there's no effect of epigenetic silencing, but then based on the qPCR analysis in S1E there is an effect. It is unclear if the authors actually quantified the sensor based on micrographs such as the one in figure 1B. If the authors properly quantified the sensor under these conditions and the effects do not match, they should provide an explanation for this. Again, ideally the authors could provide a quantification of the reporter over many individual animals and this could resolve the seeming discrepancy.

We apologize for the fact that this part of the manuscript was not written very clearly. We have re-organized this section to improve clarity, and have labelled images with % of animals in which it was observed. We hope this section is now more transparent.

2. The authors describe the mortal germline phenotype of *pid-2*, -4, -5 mutant animals and suggest it's related to the mortal germline phenotype in Piwi (*prg-1*) mutants, but don't cite a recent publication by Barucci et al, *Nat Cell Biol* 2020, that shows that *prg-1* mutant animals become progressively sterile due to the ectopic targeting of histone mRNAs. I think this is relevant work that should be cited, but in addition, it would make a lot of sense for the authors to check in their small RNA sequencing data whether they also observe small RNAs against histones. This should be a relative straightforward analysis that would further help place their observations in the context of what is known about the pathway.

This is an excellent point. We have included the citation, and we have checked 22G RNA levels from replication histone genes. We present this information in Figure S5. In our small RNA sequencing we do not detect effects on 22Gs targeting these gene classes. However, we note (and mention in the manuscript) that our small RNA sequencing was not targeted at generations that are close to sterility. Hence, we do not draw strong conclusions from the fact that we do not see an increase in histone 22G RNAs.

3. In Figure 1C, the level of Tc1 reversion in *hrde-1* single-mutants is missing for correct interpretation of the contribution of PID-2.

Thank you for spotting this omission. We have repeated the reversion assay, now including *hrde-1*, and also *prg-1* single mutants, and show the results from this new experiment. As expected, mutants that lack HRDE-1 activate Tc1 only very mildly. We also can now be stronger in our conclusion that *prg-1;pid-2* double mutants show lower Tc1 activity (basically no Tc1 activity) than *pid-2* single mutants. We interpret this as a the result of competition between PRG-1 and HRDE-1 for 22G resources. This is added to the discussion.

4. *MosSCI* and *MiniMos* alleles should be designates *xSi####* rather than *xfls####* to denote single copy integration.

Thank you for spotting this. We have corrected this.

5. In figure 7, the order in which the panels are arranged in the figure doesn't coincide with the text, it would be easier for the reader if the authors swapped 7 D,E and G,H.

We made sure Figure panels are presented in the correct order.

Thank you for submitting your revised manuscript, we have now received the reports from the three initial referees (see comments below). I am pleased to say that they overall find that their comments have been satisfactorily addressed and now support publication. Referee #3 raises two more points that should be addressed in the final revised version. Please also provide a brief point-by-point response when submitting the revised manuscript. In addition, I would like to ask you to also address a number of editorial issues that are listed in detail below. Please make any changes to the manuscript text in the attached document only using the "track changes" option. Once these remaining issues are resolved, we will be happy to formally accept the manuscript for publication.

Thank you again for giving us the chance to consider your manuscript for The EMBO Journal. I look forward to receiving your final revision. Please feel free to contact me if you have further questions regarding the revision or any of the specific points listed below.

REFEREE REPORTS

Referee #1:

In the revised version of the manuscript by Placentino et al. the authors have addressed all my major concerns about the proteomic dataset, the statistical significance of some data analysis and the rephrasing of some paragraph that render the manuscript more clear.

Referee #2:

I'm satisfied with the revisions and find the manuscript well-suited for EMBO.

Referee #3:

The authors have addressed my previous concerns and have added quantifications that help the reader interpret their data.

However, now with the rewritten first part of the manuscript there are two points that need further clarification:

1. When the authors describe the rescue constructs they claim both the fusions of PID-2 to GFP and FLAG are able to rescue, but the GFP fusions are barely able to do so. This distinction was made in the previous version but is missing here, it should be made clear again.
2. It is now clear that while both pid-2 alleles (xf23 and tm1614) have an effect on silencing of the PRG-1-dependent sensor, their effects on the epigenetically-silenced sensor (RNAe) are very different! xf23 has a noticeable effect on this epigenetic silencing (especially at 25C) but the tm1614 allele does not. This is not mentioned or discussed at all but the effects of these two alleles are so different that in my opinion this would require an attempt to rescue this phenotype as well (the authors only rescued the effect on PRG-1 dependent silencing).

Given how similar the effects of the two alleles are on small RNA populations it is difficult to understand where such a difference comes from. At the very least, this requires an attempt to explain this discrepancy.

Point-by-point response

Remaining issues raised by reviewer 3:

1. When the authors describe the rescue constructs they claim both the fusions of PID-2 to GFP and FLAG are able to rescue, but the GFP fusions are barely able to do so. This distinction was made in the previous version but is missing here, it should be made clear again.

We have corrected this by including the following sentence on page 6, line 180:

“A single-copy transgene expressing 3xFLAG-tagged Y48G1C.1, and to a lesser extent GFP-tagged Y48G1C.1, driven by its endogenous promoter and 3' UTR could rescue the 21U sensor(+) phenotype (Fig EV1B-E).”

We believe this points out the fact that the GFP tagged versions do not rescue as well sufficiently.

2. It is now clear that while both *pid-2* alleles (*xf23* and *tm1614*) have an effect on silencing of the PRG-1-dependent sensor, their effects on the epigenetically-silenced sensor (RNAe) are very different! *xf23* has a noticeable effect on this epigenetic silencing (especially at 25C) but the *tm1614* allele does not. This is not mentioned or discussed at all but the effects of these two alleles are so different that in my opinion this would require an attempt to rescue this phenotype as well (the authors only rescued the effect on PRG-1 dependent silencing). Given how similar the effects of the two alleles are on small RNA populations it is difficult to understand where such a difference comes from. At the very least, this requires an attempt to explain this discrepancy.

Our explanation for the difference is that *pid-2(tm1614)* is most likely a weaker allele than *pid-2(xf23)*. A first indication for this can be found in the fact that in both sensor reactivation assays, *pid-2(xf23)* has a stronger effect than *pid-2(tm1614)*. A second indication for this can be found in the RNAseq data the *tm1614* allele fails to show a defect in 26G RNAs (and related 22G RNAs) while the *xf23* allele does.

The issue identified by the reviewer is why the difference between the alleles is so much stronger in the sensor(RNAe) reactivation, compared to the sensor(+) reactivation assay. We believe that this difference stems from the following:

The sensor(RNAe) was crossed in into our mutants from a *prg-1* mutant background, and as we show in Figure 2, has very high 22G RNA levels. Therefore, one would expect that the sensor (RNAe) is harder to reactivate than sensor(+); indeed, even for the stronger *pid-2(xf23)* allele sensor(RNAe) reactivation is less frequent than sensor(+) reactivation. In other words, sensor(RNAe) reactivation is a more stringent assay. A relatively weak *pid-2* allele, as we assume *pid-2(tm1614)* is, could be expected to have more trouble de-silencing the sensor(RNAe). This is what we find. We note, however, that we do still observe de-silencing of sensor(RNAe) in *pid-2(tm1614)* mutants, albeit only at 25C and less frequent than in the *pid-2(xf23)* allele.

We explicitly discuss this now in the results section on page 6:

“The 21U sensor can also be in a state of RNAe: 21U sensor(RNAe). In this state, its silencing no longer depends on PRG-1, but does still rely on 22G RNAs (Ashe et al., 2012; Luteijn et al.,

2012; Shirayama et al., 2012). In contrast to the 21U sensor(+) reactivation experiment, most *pid-2* mutant animals did not reactivate the 21U sensor(RNAe) (Fig 1B), indicating that reactivation of the sensor in RNAe state is more difficult. Nonetheless, we did detect reactivation of the 21U sensor(RNAe) in some animals, most notably in *pid-2(xf23)* mutants (Fig 1B). Continuous culturing of independent cultures confirmed recurrent loss of RNAe status in *pid-2(xf23)* mutants, particularly at elevated temperature (Fig EV1F). Such loss of RNAe was much less frequent in *pid-2(tm1614)* animals (Fig EV1F). Given that the reactivation of the sensor(+) was also less effective in *pid-2(tm1614)* mutants (Fig 1B), we assume that *pid-2(tm1614)* is a weaker allele than *pid-2(xf23)*, and as such only has a very weak phenotype in the more stringent sensor(RNAe) assay, while it has an easily scored phenotype in the milder sensor(+) assay.”

We also refer back to the notion that *pid-2(tm1614)* likely is a hypomorphic allele on page 8, when we present the 26G RNA defect in *pid-2(xf23)* mutants:

“Interestingly, the strongest effect we observed on total pools of small RNA types was on 26G RNAs (Fig EV2A), and 22G RNAs produced from ERGO-1 targets were mildly reduced in *pid-2(xf23)* (Fig EV2C). Consistent with our earlier suggestion that *tm1614* may be a weaker allele of *pid-2*, *pid-2(tm1614)* mutants did not show this effect on either 26G RNAs (Fig EV2A) or associated ERGO-1 22G RNAs (Fig EV2C).”

Finally:

We did consider a rescue experiment as suggested by the reviewer, but we concluded that this may in fact be very hard to do properly. Reason for this is that loss of RNAe is only detectable in a genetic background in which PRG-1 driven silencing is hampered. If PRG-1 activity is wild-type, the non-RNAe transgene will simply be hit again by PRG-1, and be re-silenced. This is the reason why we can detect it in the *pid-2* mutants: PRG-1 activity is down, allowing loss of RNAe to be scored.

Thus, if we rescue *pid-2(xf23)* with a transgene, we are bound to not see any reactivation of the sensor(RNAe), but this would not tell us if RNAe was lost or not, since the animal is wild-type for both PRG-1 and PID-2. In other words, rescue of *pid-2* makes us blind to scoring RNAe status. We could take care of that by taking out PRG-1 genetically, but that leads to a boost in RNAe activity, as we describe in this paper (Fig 2), making it hard to predict what will happen. Any result could be explained, making it a rather weak experiment. We decided to not touch upon this issue in the manuscript, we are afraid it would convolute that part of the manuscript significantly, and unnecessarily.

Thank you again for submitting the final revised version of your manuscript for our consideration. I am pleased to inform you that we have now accepted it for publication in The EMBO Journal.

Corresponding Author Name: René F. Ketting

Manuscript Number: EMBOJ-2020-105280